# Bryophyte Flora of AlUla County (Saudi Arabia)—Distribution, Ecology, and Conservation

**DOI:** 10.3390/plants14020170

**Published:** 2025-01-09

**Authors:** Vincent Hugonnot, Florine Pépin, Jan Freedman

**Affiliations:** 1SARL Pépin Hugonnot, 43380 Blassac, France; flopepin@gmail.com; 2Royal Commission for AlUla, AlUla 43544, Saudi Arabia; j.freedman@rcu.gov.sa

**Keywords:** AlUla County, Arabian Peninsula, bryophyte conservation, checklists, endemics

## Abstract

Intensive surveys recently conducted in AlUla County (northwest Saudi Arabia) have made it possible to compile an exhaustive list of the forty-eight bryophyte species (six liverworts and forty-two mosses) known in this region and compare it with an updated checklist of bryophytes from Saudi Arabia, which now counts 31 liverworts and 135 mosses. The ecology, taxonomy, and distribution of each taxon are provided and discussed in a systematic catalog. Although particularly arid and lacking permanent watercourses, AlUla County has proven to be surprisingly rich in bryophytes. Some remarkable species are even confined here, such as *Riella affinis*, *Acaulon triquetrum*, *Microbryum rectum*, and *Syntrichia rigescens*. Two endemic species were also observed. An updated checklist of Arabian bryophyte endemics is provided. Specific recommendations regarding the conservation of bryophyte heritage are outlined.

## 1. Introduction

The largest country in southwest Asia, the Kingdom of Saudi Arabia (KSA) covers an area of approximately 2,150,000 km^2^, almost three-quarters of the Arabian Peninsula [1]. Despite the KSA generally being portrayed as a desert environment, the landscape is extremely diverse. The central and eastern regions are predominantly characterized by rocky and sandy deserts, while large mountain ranges, reaching up to 3700 m above sea level, dominate the western areas toward the Red Sea, and down toward the Yemen border. The country’s proximity to Africa, Asia, and Europe significantly influences its flora and bryoflora. With few endemic species, the bryoflora includes six floristic elements [2]. Despite containing few cosmopolitan or northern taxa, this group is rich in drought-adapted species (xerothermic Pangean or circum-Tethyan elements). The presence of various tropical species with Southeast Asian–Arabian, African–Arabian, pantropical, or Paleotropical distributions has been consistently documented [3].

With only a few species recorded in hyper-arid parts of the country, most of the bryological diversity in Saudi Arabia is concentrated in the mountainous, monsoon-influenced South-Hejaz region [2]. The hyper-arid Empty Quarter (Al-Rub’ Al-Khali Desert, located in the southeast of the KSA) has no recorded bryophyte species to date. As a result, past surveys have mainly targeted environments more conducive to bryophyte growth, such as the humid southwestern areas, leading to large portions of Saudi Arabia being neglected in bryophyte research.

Bryophyte studies in Saudi Arabia began relatively recently [4] but have steadily increased since. The first bryological checklist for the Arabian Peninsula, including Saudi Arabia, was published in 2000 [3] and was recently comprehensively revised to include all southwest Asia in 2020 [5]. The first national checklist of mosses for Saudi Arabia was published by Taha in 2019 [6], and since then, information on additional species has been recorded [7,8,9,10] or is to be published [11]. However, no comparable checklist exists for liverworts. Taking into account the information on two species previously unrecorded and published in the present paper (*Entosthodon* cf. *commutatus* and *Geheebia siccula*), the bryoflora of Saudi Arabia is currently considered to comprise 166 species (135 mosses and 31 liverworts), with no information on hornworts recorded so far.

AlUla County, located in the northwest of Saudi Arabia, south of Tabuk and approximately 200 km east of the Red Sea, covers an area of 22,561 km^2^ and has not previously been surveyed for bryophytes. During the winter of 2024, extensive surveys were conducted throughout the region, focusing on the nature reserves. In this paper, we present a checklist of bryophytes and a distribution atlas for each taxon, along with ecological data, and provide recommendations for preserving this unique and rich natural heritage.

## 2. Results

A total of forty-eight distinct taxa were observed, among which two (*Entosthodon* cf. *commutatus* and *Fissidens* cf. *arnoldii*) are taxonomic uncertainties (Table 1). Additionally, two taxa, not specifically determined (*Bryum* sp. and *Entosthodon* sp.) are recorded, but are not counted. This represents 28% of the total bryophytes recorded in Saudi Arabia.

With only six species recorded, liverworts are very rare in AlUla, with the vast majority of bryophytes being true mosses (87.5% of the taxa). All the liverworts are from the complex thalloid order Marchantiales, and no leafy liverworts were observed. Although leafy liverworts have been recorded in southwestern Saudi Arabia, the overall taxonomic profile remains quite comparable to that of the country, with 81% being mosses.

Pottiaceae are highly dominant in the bryophyte flora of AlUla, as they are throughout Saudi Arabia (Figure 3). Funariaceae and Bryaceae are also relatively well represented in both the county and the country. Some families, however, are underrepresented in AlUla County despite being fairly diverse overall (e.g., Grimmiaceae, Fissidentaceae, and some complex thalloid liverworts, such as Aytoniaceae and Ricciaceae). Additionally, six families are represented locally by only one taxon in AlUla County. Not fewer than 23 families known in Saudi Arabia do not have a single representative in AlUla County.

### 2.1. Acaulon triquetrum (Spruce) Müll. Hal. (Pottiaceae)

This is a xerothermic Pangean species that is first mentioned for the Arabian Peninsula in AlUla County [11]. This is probably a species of local occurrence in AlUla County (Figure 4) because it was observed on only one occasion at the western border of the county (Jabal Alwared). This species is only recognized when bearing sporophytes, so the collection was fully fertile, but composed of few individuals (fewer than 50). It would be interesting to locate new populations of this opportunistic species after heavy rains or an unusual prolonged period of wetness. The Arabian population is possibly attributable to *Acaulon triquetrum* var. *desertorum* (Besch.) Jelenc, but additional studies, outside the Arabian Peninsula, are required to decide if this infrataxon deserves a specific rank or not. This is an ephemeral species, also sensitive to competition from taller bryophytes, even of the same family, growing on fine-textured soil in the clearings of the Pottiaceae community.

### 2.2. Aloina rigida (Hedw.) Limpr. (Pottiaceae)

A xerothermic Pangean species widely spread on the Arabian Peninsula, it cannot be considered as a frequent species in AlUla County because it was only observed on four occasions, mostly in the west (Figure 5). The sporophytes are needed for a correct identification, but the AlUla populations were always fertile; however, they were limited spatially and with few individuals. This species was only observed in the Pottiaceae community, where it is never a dominant species.

### 2.3. Bryum dichotomum Hedw. (Bryaceae)

This is a cosmopolitan species widely spread on the Arabian Peninsula, as it is in AlUla County (Figure 6), found in a large range of geological outcrops. If *Bryum* sp. turns out to be the same species then it will be the most common taxon at the county level. It has never been observed with sporophytes nor even gametangia. However, vegetative reproduction seems to be widely favored. Many populations, especially near water sources and in shaded, often confined, habitats, show a persistent protonema, which itself is capable of vegetative reproduction (filamentous gemmae). Nevertheless, the populations are often limited to a few square centimeters of colonized area. We have adopted a broad taxonomic concept of *Bryum dichotomum*, following [12] in this regard. Ecologically, it is one of the most plastic species and apparently one of the most resistant to constraints, both hydric and mechanical (disturbances related to grazing or repeated passages of machinery). This species has been observed within six distinct bryophytic communities, some in wet habitats, others in very xeric ones, reflecting its broad adaptability.

### 2.4. Clevea spathysii (Lindenb.) Müll. Frib. (Cleveaceae)

*Clevea spathysii* is a circum-Tethyan taxon, of scattered occurrence on the Arabian Peninsula, for which information will be published separately as a new record for Saudi Arabia [11]. It was only observed in the most elevated parts of the Jabal Alward, west of the county (Figure 7), where elevation certainly results in cooler and more humid air. Nonetheless, it was only growing in fully protected micro-environments where isolation and temperatures are presumably less extreme than in exposed environments. It was bearing immature gametangia at the period of recording. Hence, this species is probably capable of producing spores. The local populations are very small, only a few tenths of thallus having been counted, making it a very vulnerable species. It grows in the Pottiaceae community.

### 2.5. Genus Crossidium (Pottiaceae) (Crossidium aberrans Holz. & E. B. Bartram, C. crassinervium (De Not.) Jur., C. deserti W. Frey & Kürschner, C. squamiferum (Viv.) Jur.)

Four taxa, all circum-Tethyan, have been observed in AlUla (Figure 8, Figure 9, Figure 10 and Figure 11). All, except for *C. deserti*, are widespread and locally abundant on the Arabian Peninsula [3,13]. Species of the genus *Crossidium* are occasionally fertile in the county. Their populations are often large, reflecting the high vitality of the taxa, as the county’s climate is favorable to species exhibiting xeropottioid syndrome [13]. The taxonomy of *Crossidium* should be re-evaluated in light of molecular tools, despite the numerous studies devoted to it [13,14,15]. In particular, the taxonomic status of *C. deserti* is questionable. Species of the genus *Crossidium* generally exhibit the same ecology, being mostly confined to the Pottiaceae community, and they are often a significant component of it, although they are also prone to colonize temporarily wetter habitats. Mixes of three species are rare (6.7% of the 74 observed populations), while those of two species are relatively frequent (at more than 20% of the 74 sites hosting *Crossidium*; Table 2). Only mixes involving *C. deserti* are absent or rare, which is obviously a consequence of its great rarity in the county.

### 2.6. Didymodon desertorum (J. Froehl.) J. A. Jiménez & M. J. Cano (Pottiaceae)

This is a circum-Tethyan taxon that is relatively widespread on the Arabian Peninsula and in AlUla County (Figure 12). It was never observed with sporophytes, even though archegonia were occasionally encountered. This species is often scarce within its populations, being limited to a few sterile stems. It is relatively frequent locally, mostly in the west of the county and, apparently, not threatened in spite of its sterility.

### 2.7. Encalypta vulgaris Hedw. (Encalyptaceae)

This species is cosmopolitan and although relatively widespread on the Arabian Peninsula (and often abundant in the Asir Mountains), it was spotted only on one occasion in AlUla County (Figure 13). It was observed with immature sporophytes bearing well-developed calyptra, allowing for a reliable identification. Only a few stems were seen in the Pottiaceae community.

### 2.8. Genus Entosthodon (Funariaceae) (E. cf. commutatus Durieu & Mont., E. duriaei Mont., and E. muhlenbergii (Turner) Fife)

*Entosthodon* cf. *commutatus* is a problematic taxon from a taxonomic perspective, quite common in AlUla County (Figure 14) and likely frequently confused with *Entosthodon attenuatus* (Dicks.) Bryhn, of which we did not observe any populations during our surveys. However, the latter species is considered as relatively common on the Arabian Peninsula. *Entosthodon* cf. *commutatus* has a long nerve, although not excurrent; its rhizoids are never cherry red but pale brown, and its capsule has a wide opening when dry, with a well-developed peristome. The populations of this species are often quite large, but sporophytes are infrequently produced, particularly in humid biotopes. Several sterile populations could not be assigned to this taxon, which likely leads to an underestimation of its actual frequency. *Entosthodon duriaei* and *E*. *muhlenbergii* are two circum-Tethyan species moderately spread on the Arabian Peninsula and more confined to the western part of AlUla County (Figure 15 and Figure 16) as compared to *E.* cf. *commutatus*. They are both moderately fertile, so they may also be underestimated. *E. muhlenbergii* is the most ecologically restricted species, being confined to the Pottiaceae community, at sites at relatively high altitudes. By contrast, the two other species are more plastic and more linked to temporarily wet tufa habitats.

### 2.9. Eucladium verticillatum (With.) Bruch & Schimp. (Pottiaceae)

This is a northern taxon that is widely spread on the Arabian Peninsula, with a concentration of records in the Asir Mountains. In AlUla, it is restricted (Figure 17) to only 11 favorable sites because of the scarcity of permanent (or nearly permanent) sources of water. It is one of the few species that is more abundant in the eastern part of the county, where permanent springs occur. This species has never been recorded with sporophytes in AlUla County. This species is generally abundant and, often, almost exclusive in its populations. This species is characteristic of the wettest tufa communities, which are permanently irrigated communities. They are also very isolated communities, like tiny islands in hyper-arid environments. The constant sterility of this species and the strong links with wet habitats suggest the relictual character of the *Eucladium* populations, which are nowadays unable to expand further and which local maintenance relies upon vegetative multiplication.

### 2.10. Fissidens cf. arnoldii R. Ruthe (Fissidentaceae)

This is a circum-Tethyan taxon, which is one of the commonest species in AlUla County (Figure 18) and is widespread, even in hyper-arid regions of the Arabian Peninsula. It has been firmly anchored in the checklist of the Arabian Peninsula since the publication of [16]. This author mentions several morphological differences from typical European *Fissidens arnoldii*, which we were able to confirm. This, combined with the occurrence of other morphologically similar species in Asia, calls for a taxonomic reappraisal of this species. It is often sterile and depauperate in many populations, but, in favorable habitats, this species is able to form relatively extended populations that are also capable of producing sporophytes. Reference [16] states “under boulders and on limestone rocks in wadis”, which fits very well what we observed in AlUla County (the Pottiaceae community). Also, it is frequently found in tufa habitats.

### 2.11. Funaria hygrometrica Hedw. (Funariaceae)

This is a sub-cosmopolitan species that is widely spread in southwestern Asia but only recorded from Kuwait, on the Arabian Peninsula [5], before its discovery in Saudi Arabia. Its information will be published separately as a new record for the country [11]. It was even more surprising that this species turned out to be very widespread (18 populations located) and abundant in suitable habitats in AlUla (Figure 19). It is often plentiful and freely fertile, with sporophytes. On certain occasions, protonematal phases, with gemmae, have also been observed. This species has obviously been neglected, and targeted research in favorable habitats, in winter, would certainly lead to a significant increase in the number of recorded populations. It grows in temporarily wet habitats, like the Riccia cavernosa community and the tufigenous *Geheebia tophacea* community.

### 2.12. Genus Geheebia (Pottiaceae)

The genus *Geheebia* has been recently extracted from the large and heterogeneous *Didymodon* on the basis of morphological and multilocus phylogenies [17]. Additionally, the taxonomic treatment of the *Didymodon tophaceus* complex has been recently challenged [18] to take into consideration considerable morphological overlap and the ITS non-monophyly of *D. tophaceus*. As a result, *G. siccula* is considered by these authors as a subspecies of *G. tophacea*. Following [7], we provisionally continue to treat this taxon as an independent species, in view of its morphological distinctiveness. *Geheebia siccula* and *G. tophacea* are two northern taxa. *Geheebia siccula* is a new record for Saudi Arabia, as it was previously mentioned in Yemen [7]. It is a rare species locally (Figure 20), which appears constantly as small populations bearing rhizoidal gemmae but no sporophytes. Ecologically, it is mostly confined to the margins of the *Geheebia tophacea* community. *Geheebia tophacea*, on the contrary, is a much more common (Figure 21) and abundant species that is widely spread in wet environments. It commonly occurs as strong populations on tufa but does not produce any sporophytes or gemmae. It is able to invade several other distinct types of tufa habitats, provided there is enough water.

### 2.13. Genus Grimmia (Grimmiaceae)

*Grimmia anodon* and *G. orbicularis* are two northern taxa with a Mediterranean character. *Grimmia anodon* is a rarity, being recorded from Jabal Alwared, at the western border of the county (Figure 22). The only other Arabian population is located at the Yemen border, at Jebel Natfa (2800 m). Only a few individuals could be spotted, despite a targeted search. By contrast, *G. orbicularis* is not only much more frequent and widespread (Figure 23) but also abundant in its populations. These two species frequently bear sporophytes, which are not produced abundantly though. *Grimmia anodon* is rather small locally compared to the size classically observed in western Europe. Its leaves are bistratose from the base to the apex, with unistratose bands appearing as striae, as seen from above. Locally, *G. orbicularis* also shows a partial bistratification of lamina. Nonetheless, the cladautoicous nature of fertile stems and cribrose and anastomosing peristome teeth leaves no doubt about the identity of the specimens. Laminar pluristratification is probably a response to the hyper-arid environment and water constraints. Both species are locally often strongly depauperate and form shorter and extended mats rather than dense cushions. They are the only species of AlUla County to grow on rocks, although G. orbicularis is also able to invade compacted soil in the vicinity of rock outcrops or boulders.

### 2.14. Gymnostomiella vernicosa (Hook. ex Harv.) M. Fleisch. (Pottiaceae)

This is a tropical species, recorded from southern Yemen and Oman, which was recently discovered in AlUla County (Figure 24), in disjunct locations far outside the tropical region. These records will be published separately [11]. This species is only present at the western fringe of the county, where rocky wadis are presumably supported by a more abundant supply of rainwater. This species is often abundant in its populations and is always sterile but constantly bears abundant gemmae. Our five specimens can be attributed to *G. vernicosa* var. *vernicosa*, as opposed to *G. vernicosa* var. *monodii* (P. de la Varde) Sérgio, as reported in Yemen and Oman. However, in the absence of a consensus on the taxonomy of this species [19,20], molecular studies would be useful. Ecologically, the observed populations grow in the minor bed of temporary rivulets (wadis), on dripping rock walls that are often incrusted with lime deposits, and within tufa bryophytic communities (with *Ptychostomum cellulare*).

### 2.15. Genus Gymnostomum (Pottiaceae)

Two species of the genus *Gymnostomum* have been identified: *G. calcareum* is a cosmopolitan taxon, whereas *G. mosis* is a circum-Tethyan one. They are both relatively widespread on the Arabian Peninsula and in AlUla County (Figure 25 and Figure 26), although *G. mosis* is much more frequent than *G. calcareum*. *G. calcareum* is less abundant in its populations as compared with *G. mosis*, which can form extensive mats over several square meters under favorable conditions. The identification of *Gymnostomum* specimens has proven to be cumbersome for two essential reasons: (1) most of the local material is sterile, sporophytes being expressed in only one population of *G. mosis*; (2) the occurrence of mixed stands with equally sterile *Gyroweisia*. *Gymnostomum gemmae* were not seen. All the gemmae-bearing material was referred to as *Gyroweisia*, and it is characterized by a relatively large size (leaves reaching 1 mm long), ovate–lanceolate leaves, mid-leaf cells that are relatively large (10 µm) and not opaque, and basal cells that are long and rectangular, often reaching mid-leaf in at least several leaves. *Gymnostomum* are of a smaller size (the best-developed leaves are smaller than 0,5 mm), with mid-leaf cells smaller (7–8 µm) and denser, and basal cells mostly quadrate or very short and rectangular. The fertile material of *Gymnostomum mosis* has a cylindrical spore capsule with a colliculate exothecium, a red rim made of transversely elongated cells, and a long rostrate operculum. The sterile material exhibits a total absence of gemmae, short leaves, and irregular stratosity of margins (occasional leaves or morphs with mostly unistratose margins). Both species are linked to tufa-forming communities.

### 2.16. Gyroweisia tenuis (Hedw.) Schimp. (Pottiaceae)

This is a northern taxon with an Atlantic–Mediterranean character. It is rather rare on the Arabian Peninsula, as it is in AlUla County (Figure 27). It is never abundant and generally poorly developed. It is locally sterile but constantly bears spindle-shaped gemmae. *Gyroweisia reflexa* (Brid.) Schimp. is recorded from the Arabian Peninsula but is not known to produce gemmae. *G. tenuis* is evidently not at its optimum and certainly has the best-developed populations in the wetter region of the peninsula.

### 2.17. Hymenostylium hildebrandtii (Müll. Hal.) R. H. Zander (Pottiaceae)

This is a Paleotropical species that is rare on the southwestern Arabian Peninsula [21] and in Makkah Province [7] and infrequent (five observations) in AlUla County (Figure 28). It occurs as small populations colonizing a few cracks in the minor beds of temporary wadis. It never produces sporophytes locally, whereas these have been observed in more southern localities. We have searched for rhizoidal gemmae in our specimens but without success. They are apparently unknown in this species. Locally, this species varies widely from a morphological point of view. The most reduced populations show tiny gametophytes with short leaves (L/l is approximately 1) to the contrary of better-developed ones that have more typical lingulate–spatulate and elongated leaves.

### 2.18. Genus Microbryum (Pottiaceae)

Three species of the genus *Microbryum* have been observed in AlUla County. Whereas *M. starckeanum*, a cosmopolitan species, is the most frequently encountered bryophyte species in AlUla County (Figure 29), the other two, both *circum-Tethyan*, are, contrastingly, among the rarest species of this county (Figure 30 and Figure 31). The sporophytes of these three species are unvaluable for specific determination. They are commonly observed, but because of frequent abortion, they do not systematically allow the spores to be checked (so a large proportion of unspecified material exists). Both *M. starckeanum* and *M. davallianum* are considered to be rather frequent at the Arabian Peninsula scale. This is not the case for *Microbryum rectum*, which has been recorded only recently as new for the entire peninsula, and its information will be published separately as such [11]. Additionally, whereas *M. starckeanum* is often present in large quantities, this is not the case for *M. rectum*, which counts only a few individuals in its unique population at the Sharaan Nature Reserve. The three *Microbryum* species are pioneer ones, invading the bare compacted substrate and being very sensitive to competition. A reflection of its abundance, *M. starckeanum* is liable to be observed in contrasting habitats, either dry (Pottiaceae community) or wet (diverse tufa communities).

### 2.19. Molendoa handelii (Schiffn.) Brinda & R. H. Zander (Pottiaceae)

This is a circum-Tethyan species previously recorded in the United Arab Emirates and Oman. Its information will be published separately as a new record for Saudi Arabia [11]. This species has a localized occurrence in AlUla County (Figure 32), and the populations do not count many individuals, as compared to other more southern locations, like the Khaybar site, where this species is more plentiful. This species has not been recorded to bear sporophytes on the Arabian Peninsula, and they are unknown to date, but foliar gemmae were observed. It is linked to the Pottiaceae community.

### 2.20. Plagiochasma rupestre (J. R. Forst. & G. Forst.) Steph. (Aytoniaceae)

This is a xerothermic Pangean taxon, which is certainly one of the most frequent and abundant marchantioids of the entire Arabian Peninsula. It is, surprisingly, of exceptional occurrence in AlUla County (Figure 33), as is the case for other complex-thallus liverworts. Climatic limitations immediately come to mind, but it should be noted that many thalloid liverworts are recorded in hyper-arid regions (with comparable levels of annual rainfall) of the peninsula [22]. The absence of the species of the large genus *Riccia* is particularly notable in this context, given that it is a genus with many xerophytic and desiccation-tolerant species. The extreme rarity of species from this phylum may stem from the history of plant colonization in the AlUla region. *Plagiochasma rupestre* is locally fertile (carpocephala observed), and only a few thalli have been observed in the rocky cracks of the *Gymnostomum mosis* community.

### 2.21. Pterygoneurum ovatum (Hedw.) Dixon (Pottiaceae)

This is a Pangean species that was previously only recorded in Kuwait, on the Arabian Peninsula [5]. The new record for Saudi Arabia will be published separately [11]. It is extremely rare in AlUla County, being only recorded from one locality (Figure 34), with very few sporophyte-bearing individuals. It was previously underlined that the local specimens were of very short stature. It was observed in the course of an open sandy wadi, in small drying depressions, and on clay–sandy material, in the Riccia cavernosa community.

### 2.22. Genus Ptychostomum (Bryaceae)

Three ecologically contrasting species of this genus are recorded in AlUla County. *Ptychostomum capillare* is a sub-cosmopolitan species of local occurrence on the Arabian Peninsula, being mostly recorded in the Asir Mountains. It was only recorded in the western mountainous part of AlUla County (Figure 35). It was previously not always distinguished from the related *P. torquescens* (Bruch & Schimp.) Ros & Mazimpaka, which also occurs in Saudi Arabia (unpublished records). In AlUla County, *P. capillare* is mostly sterile, without sporophytes, but occasionally with demonstrably dioicous gametangia. It constantly colonizes the Pottiaceae community, where it is found at a very low abundance.

*Ptychostomum cellulare* is a tropical species recorded from Oman and Yemen, on the Arabian Peninsula, which information will be published as a new record for Saudi Arabia [11]. It was recorded at a rather high number of sites (12) in the western part of the county (Figure 36). It does not produce sporophytes, but it is generally well developed from a vegetative point of view. It mostly grows not only in the *Gymnostomiella vernicosa* community but also in other tufa-forming habitats.

*Ptychostomum pseudotriquetrum* is a sub-cosmopolitan species, which has been discovered recently on the Arabian Peninsula, in southwest Yemen [23]. It is of local occurrence in AlUla County (only two observations; Figure 37) because of a lack of suitable habitats (*Eucladio verticillatae–Adiantetum capillus-veneris* Br.-BI. ex Horvatić 1934). It occurs abundantly in its two populations, covering several square meters. It is sterile, without sporophytes.

### 2.23. Riccia cavernosa Hoffm. (Ricciaceae)

This is a cosmopolitan species, with a xerothermic Pangean origin, that is widely spread on the entire peninsula, with a concentration of records along the Red Sea coast. In AlUla County, 32 populations have been recorded, which makes *R. cavernosa*, by far, the most frequent liverwort species of the county (Figure 38). It almost constantly bears sporophytes, even when individuals appear to be very depauperate and only a few millimeters in size. From a taxonomic point of view, the best developed individuals exhibit a very robust and crystalline thallus, which is somehow reminiscent of that of *R. crystallina* L., a species not mentioned on the Arabian Peninsula. The spores are typical for *Riccia cavernosa* though. The taxonomic status of the local populations would deserve a special study. It is mostly found, sometimes in large quantities, not only on sandy wadis banks after the water retreats but also, more sporadically, in other tufa-forming habitats.

### 2.24. Riella affinis M. Howe & Underw. (Riellaceae)

It is a broadly distributed, old-world amphitropical species, which information will be published as a new record for Saudi Arabia and the Arabian Peninsula [11]. Locally, this species is fully fertile, even though only immature sporophytes were observed. Two populations in two distinct ponds have been recorded to date, at the Sharaan Nature Reserve (Figure 39). One is natural and only a few square meters large (and approximately 50 cm deep during the period of the observation) at the base of a Saq sandstone canyon, along a water drainage channel. The other is of artificial origin and results from the construction of a small dam in 1970. The only associated species in the artificial pond is *Phragmites australis*, which is apparently very dynamic there.

### 2.25. Genus Syntrichia (Pottiaceae)

This genus mainly includes xerophytic species that thrive in arid and semi-arid climates. *Syntrichia caninervis* var. *caninervis* is a xerothermic Pangean species that is widely spread on the Arabian Peninsula. *Syntrichia rigescens* is a circum-Tethyan species, recorded from Israel, Jordan, and Syria [5,24]. Its information will be published as a new record for the Arabian Peninsula [11]. Both taxa are apparently rare on the Arabian Peninsula, and they are of isolated occurrence in AlUla County (Figure 40 and Figure 41) and not abundant in their populations. They are also sterile. Both are composed of diminutive individuals. However, identification is still possible based on vegetative criteria noted in [25,26]. They both colonize the Pottiaceae community.

### 2.26. Genus Targionia (Targioniaceae)

The genus *Targionia* has been the subject of detailed studies on the Arabian Peninsula [27,28], which have led to the description of a new endemic taxon, *T. hypophylla* subsp. *linealis*. Despite targeted searches, we were unable to observe this subspecies in AlUla County. However, the types of subspecies, along with *T. lorbeeriana*, were recorded. Both species are widespread on the Arabian Peninsula, with a concentration of records along the Red Sea escarpment. In the county, both species are very rare (Figure 42 and Figure 43), though *T. hypophylla* is more common than *T. lorbeeriana*. Both species occasionally produce sporophytes, but their populations are often extremely low in number. Clearly, the genus *Targionia* deserves a more in-depth taxonomic study using molecular tools. Both species are mostly found within the Pottiaceae community, but *T. hypophylla* is also able to spread into the tufa-forming *Gymnostomum mosis* community.

### 2.27. Timmiella barbuloides (Brid.) Mönk. (Pottiaceae)

This is one of the few species that is widespread on the Arabian Peninsula (particularly in western Hijaz and Asir) and in western AlUla County (Figure 44). This is a circum-Tethyan taxon that is locally abundant but which sporophytes are of exceptional occurrence in AlUla County. Sexual patterns have been verified (this species is paroicous) in each of the few fertile gatherings, but all the sterile material was also referred to this species. The only other species of this genus, *T. anomala* (Bruch & Schimp.) Limpr., is only recorded in Kuwait, on the entire Arabian Peninsula. This species occurs mostly in the *Gymnostomum mosis* community, in other tufa-forming ones, and, more rarely, in the Pottiaceae community, which certainly reflect its affinity toward temporarily wet substrates.

### 2.28. Genus Tortula (Pottiaceae)

Four species belonging to the genus *Tortula* have been recorded in this county. These species are more or less widespread across the Arabian Peninsula, but all are more abundant in mountainous regions, such as Asir, Hijaz, or the cuesta of Jabal Tuwayq. In AlUla County, this genus is of particular importance because of its abundance. Although *Tortula atrovirens* is one of the most common species (Figure 45), *T. inermis* (Figure 46), *T. muralis* (Figure 47), and *T. mucronifera* (Figure 48) are also relatively frequent in this county. All the *Tortula* species show a strong affinity for the most xeric habitats, notably those in the Pottiaceae community. Although these species are capable of producing sporophytes, they often abort early in the season because of water stress. Fortunately, all of them can be easily identified, even in the absence of a spore capsule.

### 2.29. Trichostomopsis australasiae (Hook. & Grev.) H. Rob. (Pottiaceae)

This is one of the most common species in AlUla County and on the Arabian Peninsula. Like many other xerophytic species, it occurs principally in Jabal Tuwayq and the Asir Mountains. Most of its AlUla populations are concentrated in the western part of the county (Figure 49). It belongs to a complex genus that is currently considered to contain only two species, the other one, *T. umbrosa* (Müll. Hal.) H. Robins, has a much more restricted range on the peninsula, being only recorded in Saudi Arabia. *T. australasiae* is a polymorphic species that is surprisingly stenotypical in AlUla County, perhaps reflecting the overall aridity of this region. It has never been observed with sporophytes locally, whereas they seem to be rare globally. It grows in the Pottiaceae community and in various tufa-forming habitats.

### 2.30. Vinealobryum vineale (Brid.) R. H. Zander (Pottiaceae)

It is of scattered occurrence on the Arabian Peninsula, as it is in AlUla County (Figure 50). This species is found in small populations made up of clustered individuals, which never produce sporophytes. They are easily spotted within mixed Pottiaceae species because of the frequent rusty tinge of their leaves. They are, nevertheless, easily determined owing to the occurrence of a small translucent window, made of smooth cells, in the apical leaf canal. We have verified the constant occurrence of quadrate dorsal costal cells because of the potential taxonomical confusion with the related *Vinealobryum insulanum* (De Not.) R. H. Zander, which has elongated dorsal cells. *V. vineale* grows in the *Gymnostomum mosis* community or in the Pottiaceae community.

### 2.31. Weissia condensa (Voit) Lindb. (Pottiaceae)

This is a relatively common species in mountainous regions of the peninsula. In AlUla County, it is of moderately frequent occurrence (Figure 51). This species belongs to a particularly complex and large genus, which would benefit from a taxonomic revision at the Arabian Peninsula scale or at a larger scale. Moreover, it does not locally produce sporophytes, although they are known elsewhere on the peninsula. Our specimens are attributed to this species, mainly based on the presence of a relatively short leaf with an obtuse apex, featuring a wide nerve (more than 70 µm wide at the base). *Weissia latiuscula* Müll. Hal. was mistakenly reported on the peninsula, as the specimens so named correspond to a short-leaved form of *W. condensa* [5]. *W. condensa* is mainly confined to the Pottiaceae community.

## 3. Discussion

We now have an updated checklist of the bryoflora of Saudi Arabia (Appendix A), as well as a checklist of the bryoflora of AlUla County (Table 1). In total, 28% of Saudi Arabia’s bryophyte flora is recorded in AlUla County, which covers just over 1% of the country’s area. Forty-eight bryophyte species (six liverworts and forty-two mosses) are now known in AlUla County as compared to the 31 liverworts and 135 mosses so far recorded in Saudi Arabia. This means not only that in a regionally hyper-arid region, AlUla County is certainly rich in bryophyte taxa but also that the phyla are in need of a systematic recording method at the country or, even better, at the Arabian Peninsula scale. The uniqueness of AlUla County is highlighted by the presence of four species that are not recorded anywhere else on the Arabian Peninsula: *Riella affinis*, *Acaulon triquetrum*, *Microbryum rectum*, and *Syntrichia rigescens*.

Advances in taxonomic knowledge have led many species initially considered as endemic to the Arabian Peninsula to be synonymized with more widely distributed taxa. The current number of endemic taxa on the Arabian Peninsula is difficult to determine without up-to-date research, but recent studies have generally shown a trend of decreasing numbers of species considered endemic because of these taxonomic revisions (Table 3). As a result, only 12 taxa are currently considered as endemic on the Arabian Peninsula, whereas two of them (*Crossidium deserti* and *Tortula mucronifera*) have been recorded in AlUla County.

Several species (*Entosthodon* cf. *commutatus* and *Fissidens* cf. *arnoldii*) will require a detailed taxonomic study aimed at clarifying their identity. To achieve this, the use of molecular tools will be essential, given the limited morphological characteristics available in these simplified organisms [40,41]. This will be the subject of forthcoming publications.

Interestingly, some species that are well adapted to arid, or even hyper-arid, climates are extremely rare or completely absent in AlUla, despite being characteristic of other arid regions of the Arabian Peninsula (e.g., *Didymodon acutus* and *Riccia* spp.). On the other hand, some species that are relatively common in this county (e.g., *Fissidens* cf. *arnoldii*, *Microbryum starckeanum*, and *Riccia cavernosa*) are not necessarily more xerophytic than those that are absent. A similar observation applies to tropical species: some western wadis support communities of hygrophytic species with tropical affinities. These habitats are often completely devoid of xerophytic species, even at their margins, which are often shaded or temporarily wet, suggesting that more mesophytic species might be expected. Therefore, although the extremely harsh regional climate is a significant factor, it does not fully explain the distinctive floristic composition of AlUla County. Other factors must be considered, though they are harder to identify through simple field observations.

Floral migrations due to climate change have likely played an important role. The presence of tropical species suggests a shared flora during the Tertiary, when the Arabian Peninsula may have acted as a migration bridge between Asia and Africa [42]. The aridification of the peninsula in the Late Tertiary forced much of this tropical flora to migrate southward to elevated mountain ranges affected by monsoons, where these populations are now highly fragmented and isolated [43]. This study demonstrates that tropical taxa can also survive in arid regions as sparse and sterile populations within highly localized microhabitats, far from areas influenced by maritime monsoon mists and fogs. However, the scarcity of xerophytic species must be explained differently. Recolonization by xerophytic species may be extremely slow, and some species may not yet have had enough time to establish locally. Additionally, bryophytes, like the vascular flora, are subject to anthropogenic pressures, particularly overgrazing, which has resulted in physical damage, loss of biomass, soil compaction, microclimate alteration, erosion, etc., which, in turn, lead to significant changes to vegetation and the disappearance of many taxa.

## 4. Conservation

No conservation actions specifically targeting bryophytes have yet been undertaken in AlUla County, or even in Saudi Arabia. Yet, the conservation of bryophytes in arid environments is of the utmost importance, despite their small size and inconspicuous appearance. These plants play a crucial role in hyper-arid ecosystems, contributing to soil formation, water retention, and surface stabilization, thereby preventing erosion [44,45]. Bryophytes also serve as microhabitats for many microorganisms and insects, enriching the biodiversity of these fragile environments [46]. Furthermore, as pioneer species, they are often the first to colonize degraded areas, facilitating the establishment of other vegetation [47]. In arid environments, bryophytes face extreme climatic conditions and have developed remarkable adaptations to withstand drought and high temperatures [48,49]. Their preservation is essential, as they are sensitive indicators of ecosystem health [50]. In AlUla County, developments of all kinds, even those of limited scope and without significant artificial alteration, as well as illegal dumping, construction of shelters, etc., can directly threaten bryophyte assemblages by simply destroying or altering the ecological characteristics of microhabitats.

The case of *Riella affinis* is particular and deserves a separate discussion. The populations of this species are certainly among the most important to conserve in AlUla County because of their international rarity, ecological interest, and biogeographical significance. The two water bodies that this species inhabits are potentially threatened, either by changes in its water supply or by a breach of the dam, and should, therefore, be monitored closely. A detailed study of habitat conditions and monitoring of the populations should be very useful to uncover tendencies and evaluate the relevancy of carrying out active management actions. The gradual silting caused by the natural accumulation of sediment could lead to the infilling of the waterbody and its invasion by *Phragmites australis*, which is already well established.

The conservation of tufa ecosystems in arid environments is critical because of their unique ecological and hydrological roles. In arid regions, tufa ecosystems provide vital microhabitats with relatively stable moisture conditions, offering refuges for species that are poorly adapted to the broader hyper-arid environment [51]. These include several specialized, desiccation-sensitive bryophytes of tropical affinities (*Gymnostomiella vernicosa*, *Hymenostylium hildebrandtii*, and *Ptychostomum cellulare*) or northern taxa (*Ptychostomum pseudotriquetrum*, *Eucladium verticillatum*, etc.). Tufa ecosystems are sensitive to changes in hydrology, nutrient inputs, and sediment accumulation [52]. Human activities, such as groundwater extraction, overgrazing, and land-use changes, pose significant threats to these fragile systems, often resulting in reduced water flow and increased sedimentation, which can alter the habitat and lead to the loss of specialized species [53]. Conservation strategies should focus on maintaining natural hydrological regimes, reducing human-induced nutrient inputs, and controlling activities that accelerate sediment accumulation to preserve the ecological integrity of these unique habitats [54].

Bryophytes in arid environments are highly sensitive to grazing pressures because of their small size, slow growth, and reliance on surface moisture for survival. Grazing by livestock (mostly camels, goats, and donkeys) can directly damage bryophyte colonies by trampling, which physically destroys or dislodges the plants, and indirectly by altering microhabitats through soil compaction and changes in the surface structure [55]. The Pottiaceae community, which is the floristically the richest of AlUla County, requires soil substrates embedded in rocks and showing signs of temporary stabilization. The movement of rocks and mechanical disturbances resulting from the passage of livestock can lead to destabilization, which is detrimental to these species. This pressure can lead to reduced bryophyte cover, decreased species diversity, and even local extinctions in heavily grazed areas [46]. This is also true for saxicolous assemblages, which are of very restricted occurrence in this county and are strictly dependent upon the arrangement of the stones, which creates environments that are relatively less exposed to intense solar radiation, probably allowing for spore germination and protonema colonization. On the other hand, the pioneer species could benefit from the repeated passage of animals and soil compaction, but these disturbances must remain low in intensity and short in duration. The sensitivity of bryophytes to grazing highlights the need for careful management of grazing intensity and timing in arid ecosystems to prevent irreversible damage. Strategies, such as rotational grazing, limiting livestock numbers, and establishing grazing exclusion zones, can help to mitigate the impacts of grazing and support bryophyte conservation [56].

The example of the Sharaan Nature Reserve (Figure 52) is particularly instructive. In 2021, a fence was erected around the entire nature reserve. This has protected the floral species from grazing by camels and goats and subsequently allowed for the reappearance of annual and perennial tracheophyte communities that no longer exist elsewhere. Under the cover of these species, in relatively cool microhabitats sheltered from direct solar radiation, several interesting Pottiaceae can be found, including the rare *Microbryum rectum*.

Climate change poses a significant threat to the conservation of bryophytes in arid regions. Bryophytes are highly sensitive to changes in temperature and moisture because of their poikilohydric nature—they lack specialized tissues for water retention and are directly influenced by their surrounding environment [45]. Rising temperatures and increased frequency of drought events, driven by climate change, exacerbate the already harsh conditions of arid ecosystems, making it more challenging for bryophytes to survive and reproduce [57]. Abortive embryonic sporophytes have often been observed and highlight the fact that many species do not complete their development cycle every year (e.g., many Pottiaceae, *Entosthodon*, and *Bryum*). A minimal vegetation cover would certainly help to promote sexual reproduction and the full development of sporophytes, but such a cover remains extremely limited because of widespread overgrazing.

## 5. Materials and Methods

### 5.1. Study Site

AlUla County is situated in the northwestern part of the Arabian Peninsula (Figure 53) and falls within the Saharo–Sindian regional zone [21,58], which is generally characterized by an arid to hyper-arid climate, receiving less than 50 mm of annual rainfall.

The geology of AlUla is very diverse for such a small region. It is a part of the Arabian Shield, containing a wide variety of Precambrian igneous and metamorphic rocks, along with Cambrian and Early Ordovician sandstones [59]. The central and northern parts of AlUla are dominated by the Siq, Quweira, and Saq Sandstone Formations, which are mainly composed of fine- to medium-grained quartz [59]. Tectonic activity dating from around 10 million to 500,000 years ago, related to the rifting of the Red Sea, led to the formation of several extensive basaltic lava fields, locally known as harrats [59,60,61]. More recent alluvial deposits are found throughout the region, especially in valleys (wadis), ranging from large water-transported boulders to gravels and fine-grained wind-blown sand [62].

A diverse geodiversity results in a variety of habitats, which, in turn, lead to increased biodiversity [63,64,65]. In AlUla, the erosion of the different rock types, combined with limited vegetation cover and low annual rainfall, has created a wide range of habitats. The crystalline igneous and metamorphic rocks in the south erode differently compared to the northern sandstones, giving rise to distinct landscapes and soil types. This rich geodiversity has contributed to the establishment of five nature reserves, covering over 50% of the county, each with its unique characteristics shaped by the underlying geology: Harrat Uwayrid, Harrat AlZabin, Wadi Nakhlah, Sharaan, and AlGharameel Nature Reserves.

The climate across most of northern Arabia features very hot, dry summers and from cool to warm winters. Precipitation mainly comes from cyclonic depressions during winter and spring, originating in the Atlantic or Mediterranean. Rainfall is irregular and unpredictable in timing, quantity, and distribution, often falling as heavy, localized showers. Some areas may receive almost no rain in certain years. Meteorological data from the AlUla station provide a good representation of the regional climate (Figure 54), with a mean annual rainfall of 61 mm and a mean annual temperature of 24.6 °C.

In AlUla County, the main types of tracheophyte vegetation units are the following [66,67]:*Lycium shawi*–*Ephedra foliata* shrubland of the runnels and wadis of the Hijaz Mountains;Vegetation of the sandstone canyons and sandy gullies (*Ficus salicifolia*, *F. palmata*, *Abutilon bidentatum*, etc.);A sparse mixed herb and shrub community of the scree slopes (*Ochradenus baccatus*, *Fagonia bruguieri*, *Forsskaolea tenacissima*, and *Tetraena simplex*) housing crack micro-communities dominated by *Parietaria alsinifolia*, *Isatis lusitanica*, and *Hemionitis pteridioides*);The *Retama raetam*–*Artemisia sieberi* open shrubland of the high Hijaz Mountains, well developed in the Jabal AlWard from 1500 to 2000 m;Palm groves housing rich assemblages of ruderal communities (*Aizoon canariense*, *Lysimachia arvensis*, *Calendula tripterocarpa*, *Malva parviflora*, *Bromus tectorum*, etc.);Seasonal pools with *Juncus rigidus*, *Typha* sp., and *Mentha longifolia* and mainly Characeae species and green Algae;Seepage with *Adiantum capillus veneris*.

### 5.2. Surveys and Nomenclature

The surveys were conducted from 13 January to 27 February 2024, with 488 selected sites surveyed (Figure 55). The selection of the sites was informed by the examination of geological and topographical maps to identify potential sources of moisture, such as wadis, springs, and wetlands. An additional site selection criterion was accessibility, based on the presence of trails within a clear and safe walking distance of less than five kilometers from a road or a track. Local rangers’ information regarding safe access for prospecting and the presence of potential habitats for bryophytes was included in the site selection. All the sites meeting the aforementioned selection criteria were visited during the field session.

A total of 207 bryosociological relevés were systematically taken using the traditional approach [68] and the adapted abundance–dominance scale [69], which corresponds to 842 floristic data. Associated species cited in the present work are those sharing the same microhabitat. Classical ecological parameters were sampled (e.g., exposure, inclination, cover of bryophytes, and soil depth, where relevant). The description of the recognized bryophyte communities is to be published separately [70] (Table 4). The host communities of each bryophyte species were determined retrospectively based on bryosociological survey tables; the frequency of the occurrence of each species is presented as a frequency histogram inset within each distribution map. The tracheophytic vegetation description is mostly derived from the Inventory and Vegetation of AlUla Flora Project [66,67]. Associated vascular plants are systematically noted, with a distinction made between those growing in the same microhabitat (e.g., a wet fissure or under an overhang) and those growing in the vicinity under distinct ecological conditions.

The definition of the Arabian Peninsula follows [3] and, therefore, excludes Iraq and Jordan. The taxonomy and nomenclature of the bryophytes mainly follow [5], except for the 13 names indicated in Table 5, for which specific references are provided. For vascular plants, we use [67]. The vouchers are currently kept in the private herbarium of Vincent Hugonnot and Florine Pépin and will be transferred to the Royal Commission for AlUla’s (RCU) collection in early 2025.

Several species are taxonomically problematic and are currently under reassessment. Such taxa have been named using up-to-date treatments, i.e., [5], including *Fissidens* cf. *arnoldii* R. Ruthe and *Entosthodon* cf. *commutatus* Durieu & Mont. The latter is not mentioned in [5]. *Bryum* sp. mostly corresponds to the putative *Bryum dichotomum* Hedw., but in the absence of sporophytes, axillary bulbils, and tubers, this determination remains doubtful. *Entosthodon* sp. corresponds either to *E.* cf. *commutatus* Durieu & Mont. or to *E. duriaei* Mont.

Each species is examined in terms of its biogeographical affinities, distribution on the Arabian Peninsula and in AlUla County, frequency, fertility, morphological plasticity or polymorphism, and its taxonomy. The biogeographical elements are mostly extracted from [13]. The rarity of each species is difficult to assess because of the relatively incomplete floristic data available. However, based on the work of [3] and more recent publications, it is possible to propose a frequency quantification using a very simple ordinal scale (C: relatively common species, probably often underestimated; R: relatively rare species; TR: seemingly very rare species and likely not underestimated). The base maps have been adapted from those available at this address: https://www.floodmap.net (accessed on 12 November 2024), © OpenStreetMap contributors. We have compiled all the citations from the literature to create an updated checklist of the bryoflora of Saudi Arabia, as provided in the Appendix A.

The heritage value of each taxon is assessed somewhat subjectively (in the absence of a reference document), considering its rarity within AlUla County, rarity across the Arabian Peninsula, biogeographical affinities, and its degree of ecological specialization. The most remarkable species are ranked using an original three-level ordinal scale (x: species with high conservational importance; xx: species with very high conservational importance; xxx: species with exceptional conservational importance). The conservation of the bryophyte heritage and the most remarkable or potentially threatened species are described in the discussion, taking into account species assemblages and respective communities.

## Figures and Tables

**Figure 1 plants-14-00170-f001:**
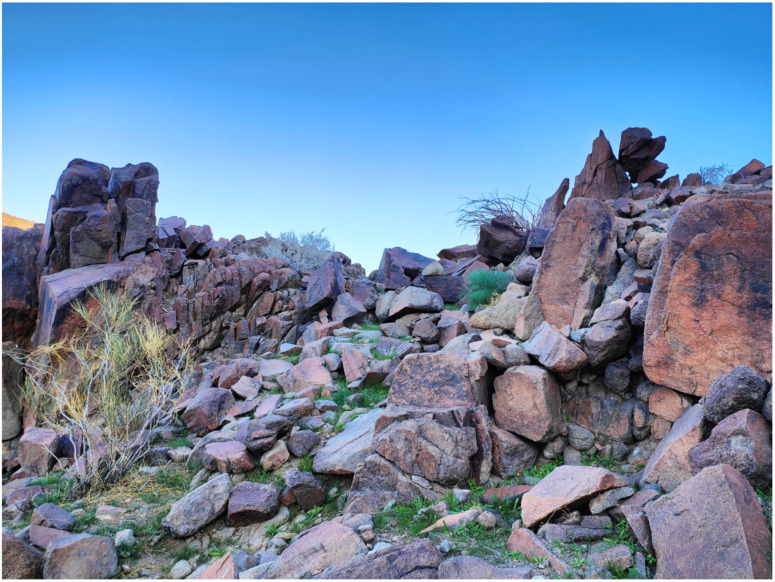
Granite scree hosting the Pottiaceae community, with four species of the genus *Crossidium* (community code 3, see Section 5).

**Figure 2 plants-14-00170-f002:**
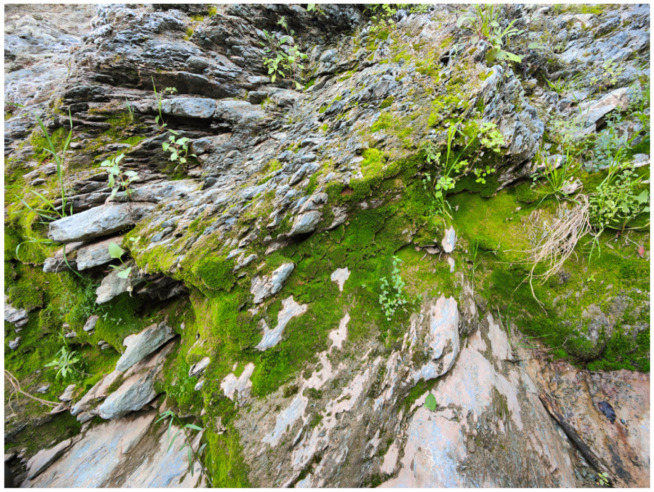
*Gymnostomum mosis* tufa community (community code 10, see Section 5).

**Figure 3 plants-14-00170-f003:**
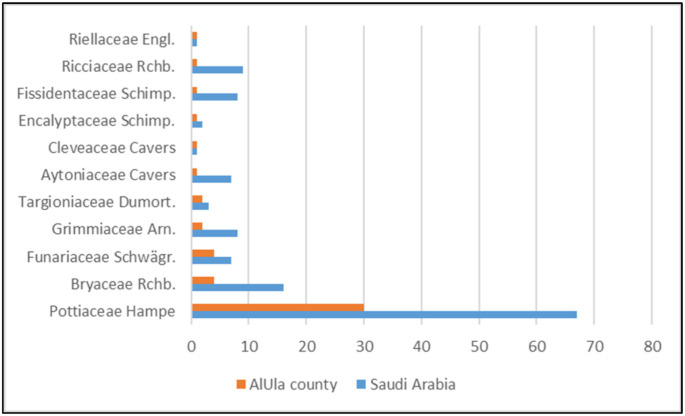
Family representation within the bryoflora of AlUla County as compared with Saudi Arabia.

**Figure 4 plants-14-00170-f004:**
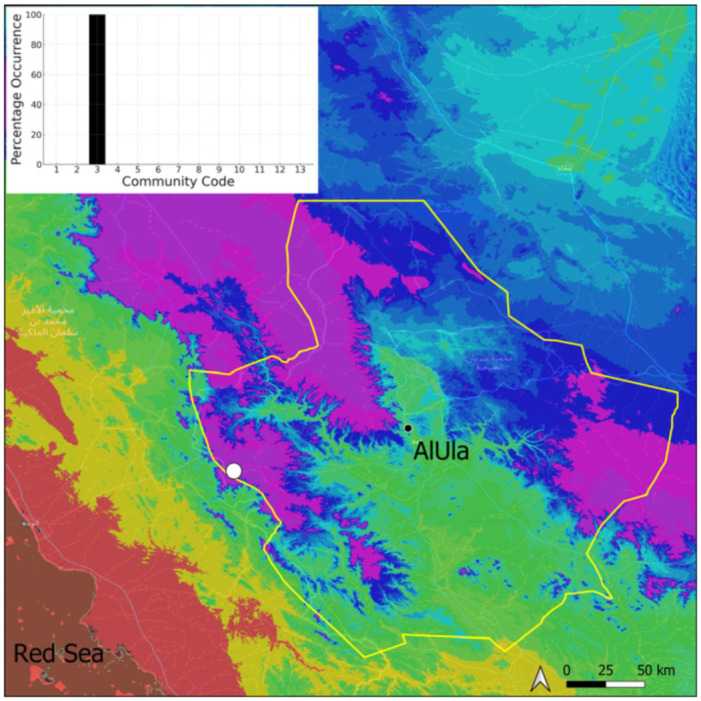
Distribution and habitat (inset) of *Acaulon triquetrum* (Spruce) Müll. Hal. in AlUla County.

**Figure 5 plants-14-00170-f005:**
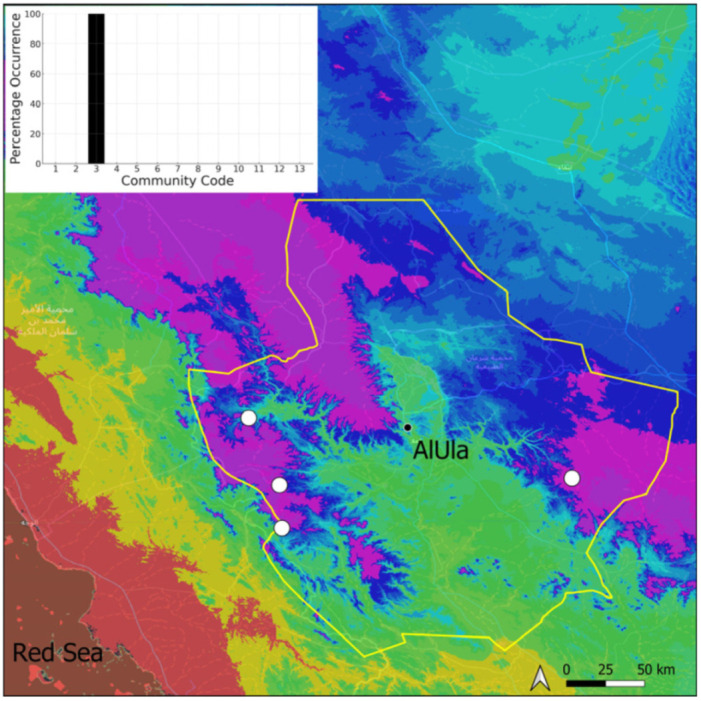
Distribution and habitat (inset) of *Aloina rigida* (Hedw.) Limpr. in AlUla County.

**Figure 6 plants-14-00170-f006:**
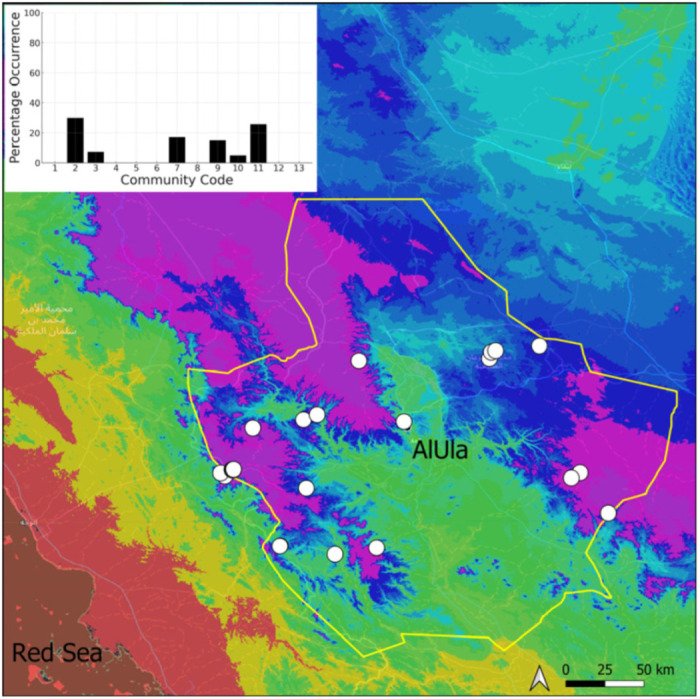
Distribution and habitats (inset) of *Bryum dichotomum* Hedw. in AlUla County.

**Figure 7 plants-14-00170-f007:**
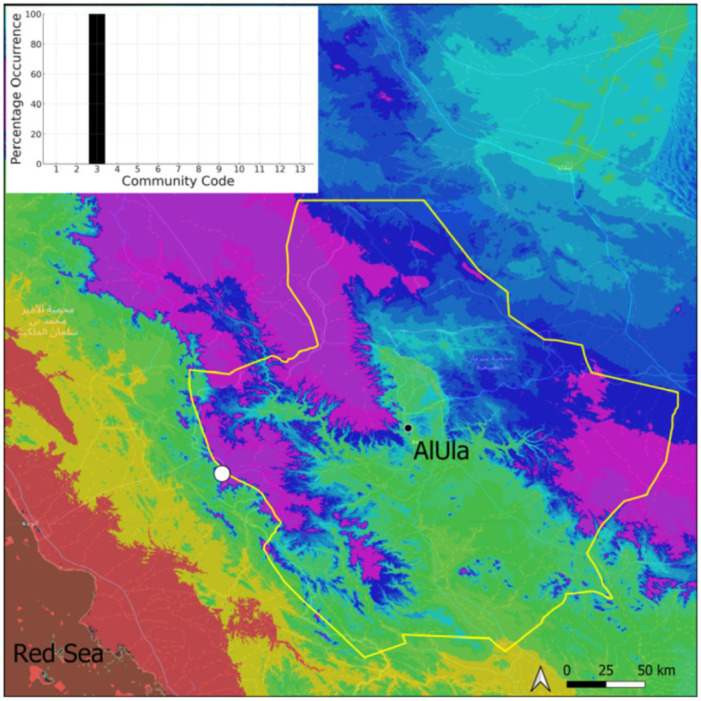
Distribution and habitat (inset) of *Clevea spathysii* (Lindenb.) Müll. Frib. in AlUla County.

**Figure 8 plants-14-00170-f008:**
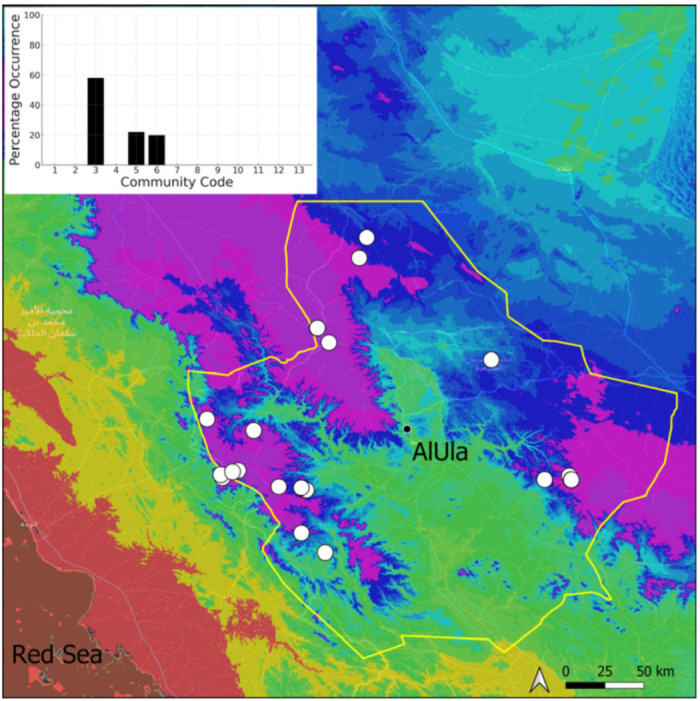
Distribution and habitats (inset) of *Crossidium aberrans* Holz. & E. B. Bartram in AlUla County.

**Figure 9 plants-14-00170-f009:**
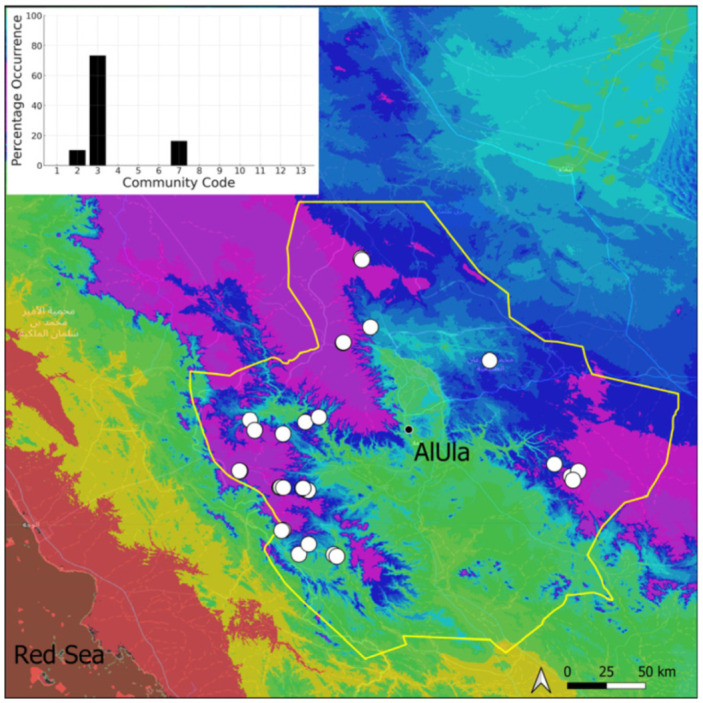
Distribution and habitats (inset) of *Crossidium crassinervium* (De Not.) Jur. in AlUla County.

**Figure 10 plants-14-00170-f010:**
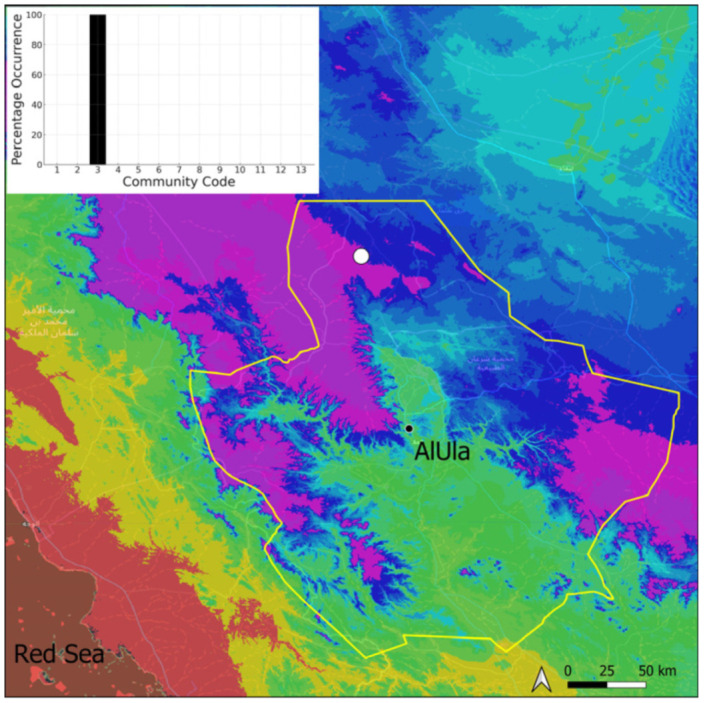
Distribution and habitat (inset) of *Crossidium deserti* W. Frey & Kürschner in AlUla County.

**Figure 11 plants-14-00170-f011:**
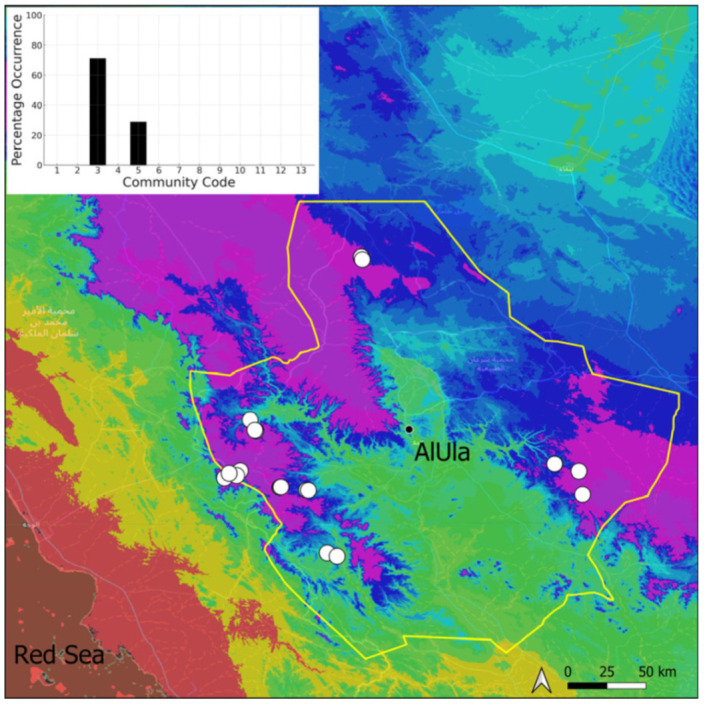
Distribution and habitats (inset) of *Crossidium squamiferum* (Viv.) Jur. in AlUla County.

**Figure 12 plants-14-00170-f012:**
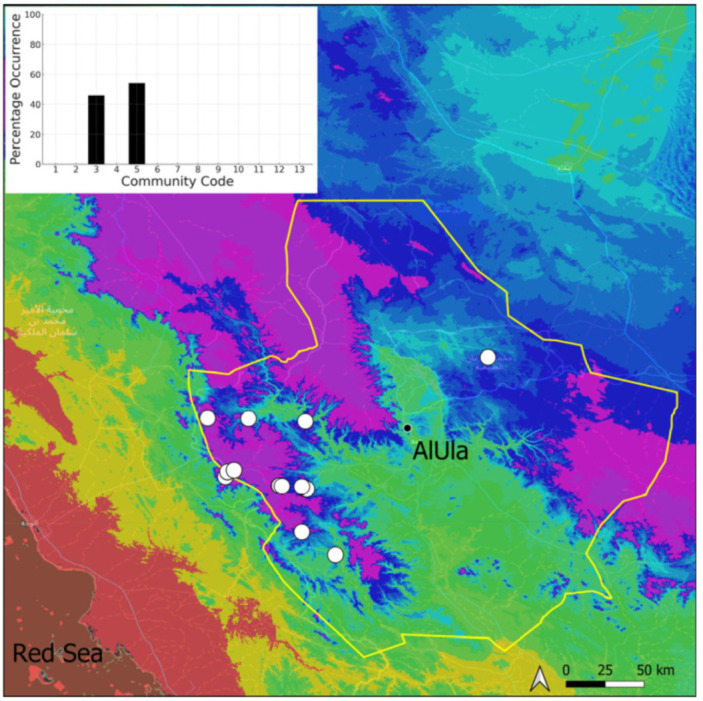
Distribution and habitats (inset) of *Didymodon desertorum* (J. Froehl.) J. A. Jiménez & M. J. Cano in AlUla County.

**Figure 13 plants-14-00170-f013:**
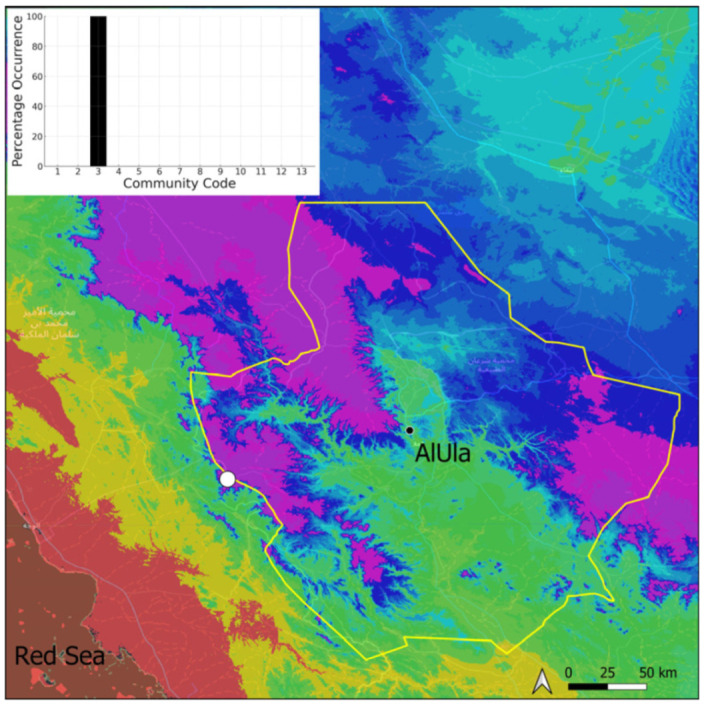
Distribution and habitat (inset) of *Encalypta vulgaris* Hedw. in AlUla County.

**Figure 14 plants-14-00170-f014:**
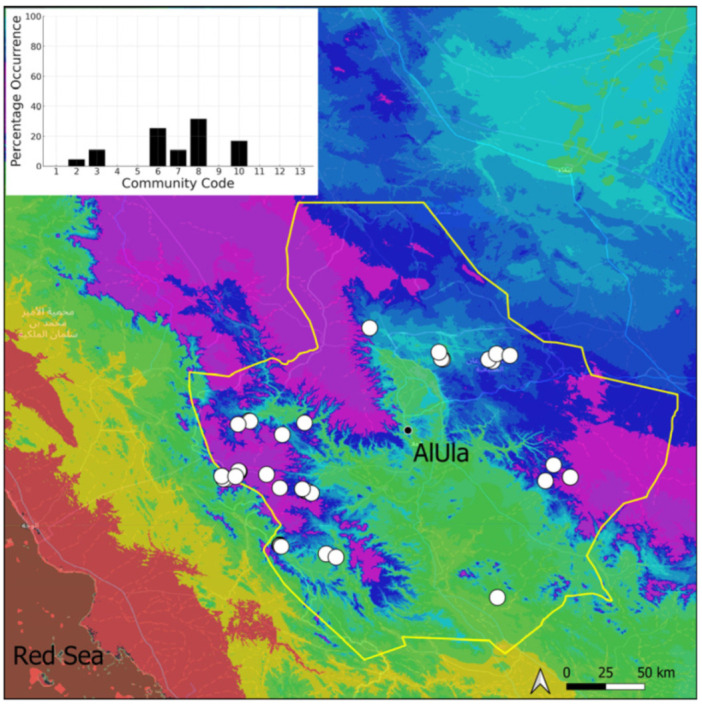
Distribution and habitats (inset) of *Entosthodon* cf. *commutatus* Durieu & Mont. in AlUla County.

**Figure 15 plants-14-00170-f015:**
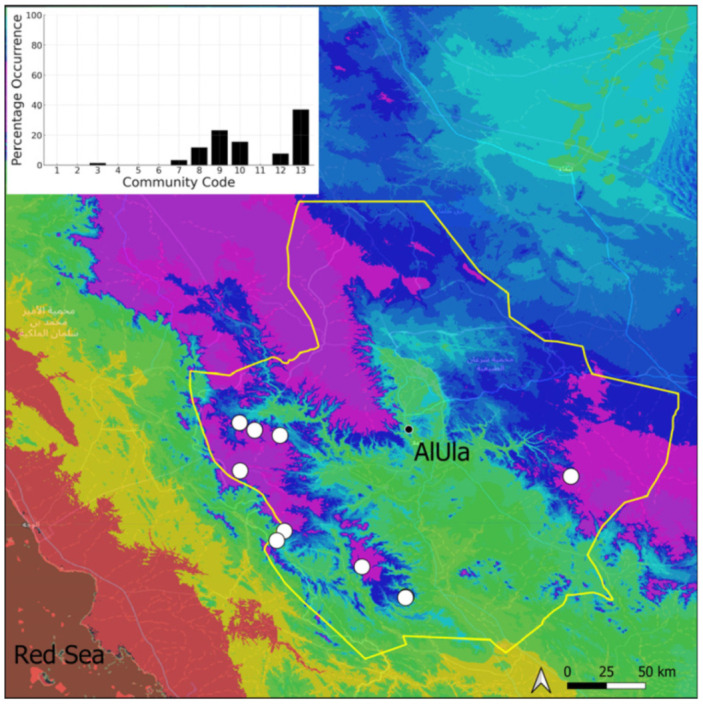
Distribution and habitats (inset) of *Entosthodon duriaei* Mont. in AlUla County.

**Figure 16 plants-14-00170-f016:**
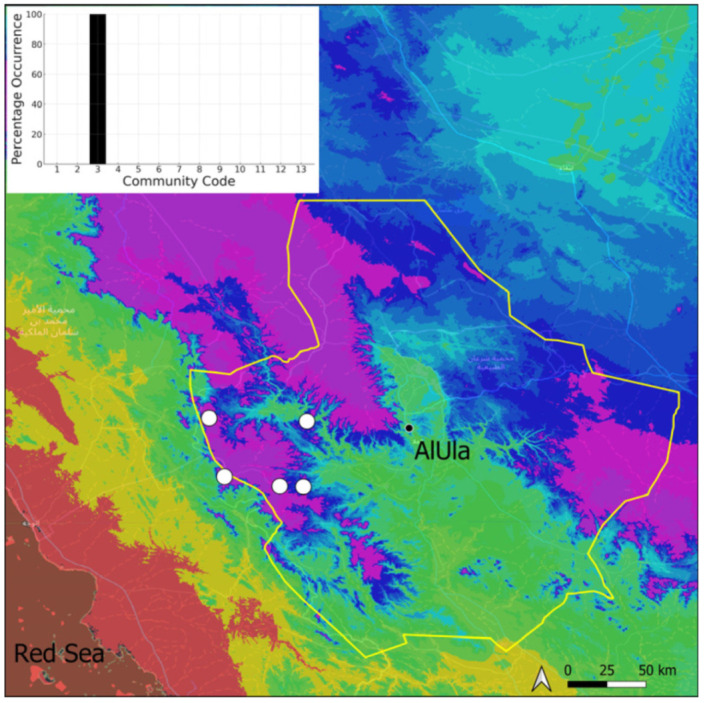
Distribution and habitat (inset) of *Entosthodon muhlenbergii* (Turner) Fife in AlUla County.

**Figure 17 plants-14-00170-f017:**
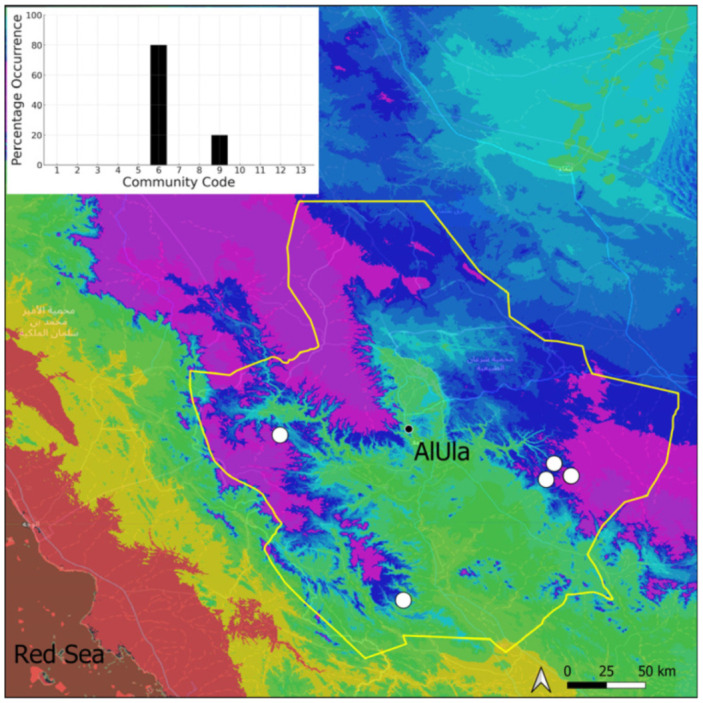
Distribution and habitats (inset) of *Eucladium verticillatum* (With.) Bruch & Schimp. in AlUla County.

**Figure 18 plants-14-00170-f018:**
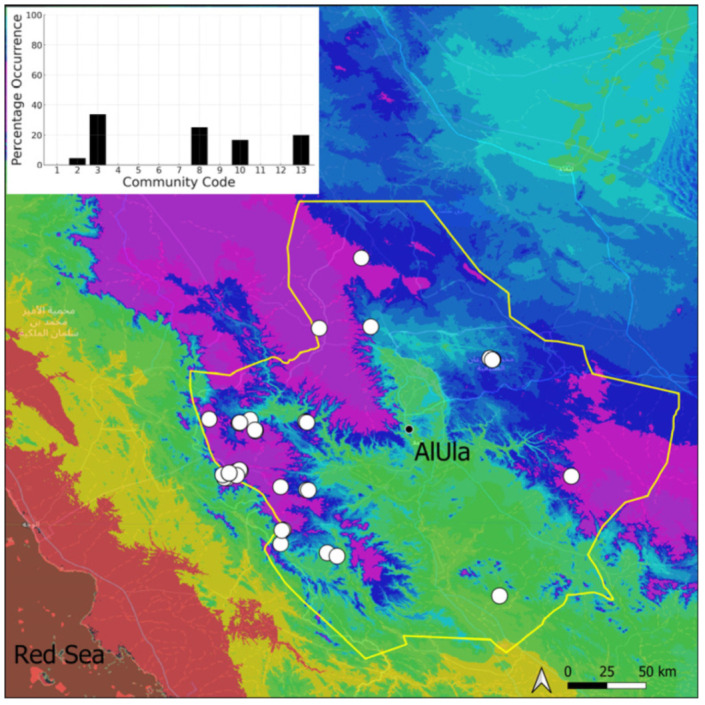
Distribution and habitats (inset) of *Fissidens* cf. *arnoldii* R. Ruthe in AlUla County.

**Figure 19 plants-14-00170-f019:**
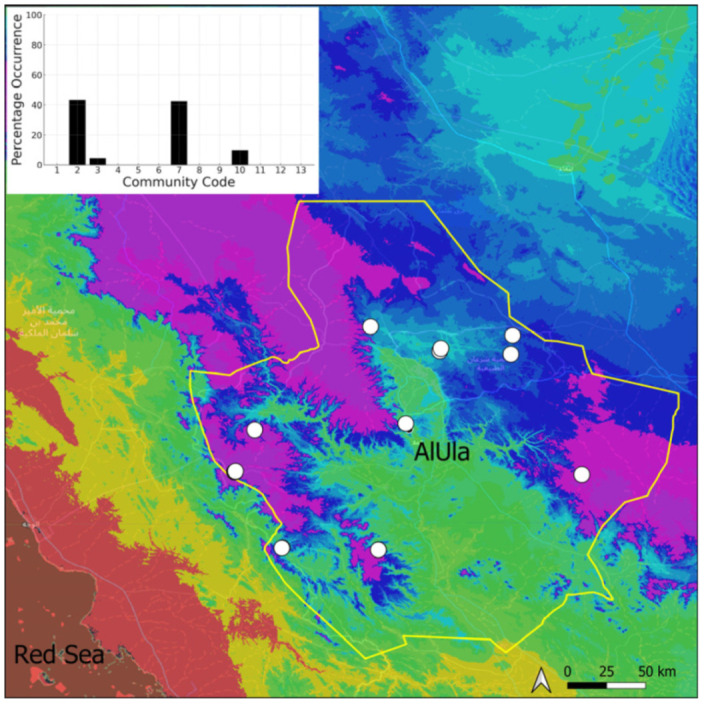
Distribution and habitats (inset) of *Funaria hygrometrica* Hedw. in AlUla County.

**Figure 20 plants-14-00170-f020:**
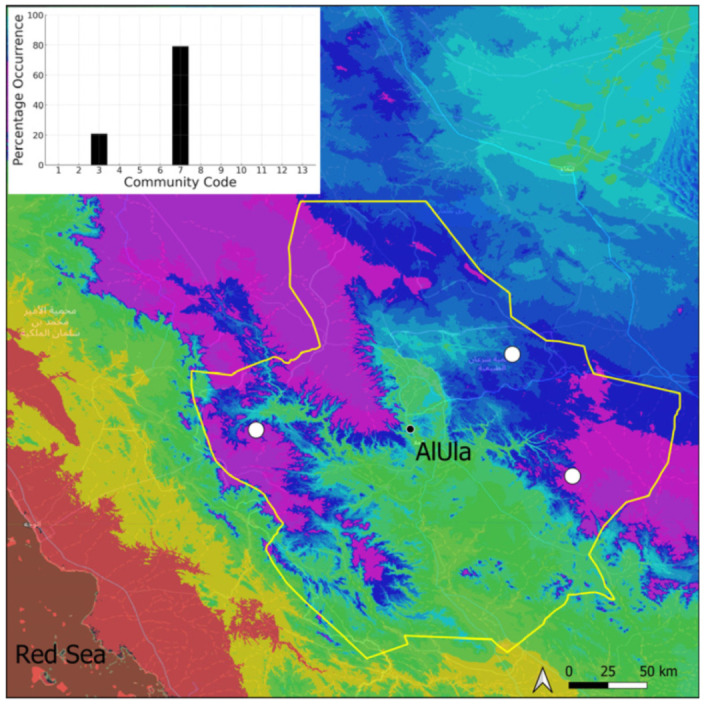
Distribution and habitats (inset) of *Geheebia siccula* (M. J. Cano, Ros, García-Zam. & J. Guerra) J. A. Jiménez & M. J. Cano in AlUla County.

**Figure 21 plants-14-00170-f021:**
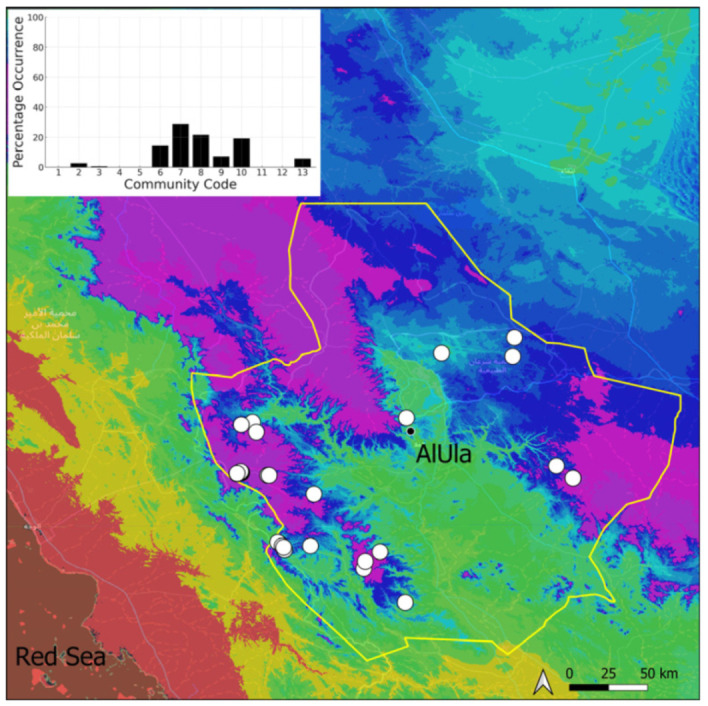
Distribution and habitats (inset) of *Geheebia tophacea* (Brid.) R. H. Zander in AlUla County.

**Figure 22 plants-14-00170-f022:**
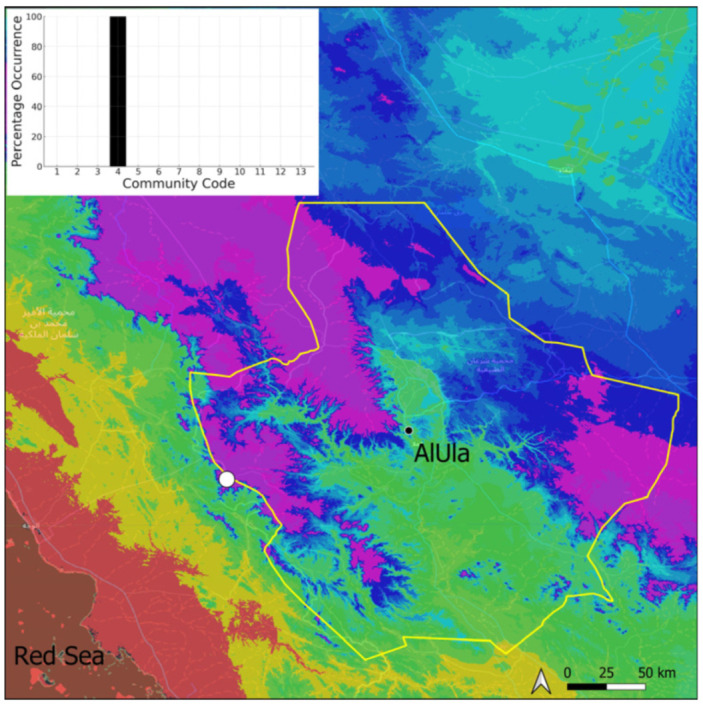
Distribution and habitat (inset) of *Grimmia anodon* Bruch & Schimp. in AlUla County.

**Figure 23 plants-14-00170-f023:**
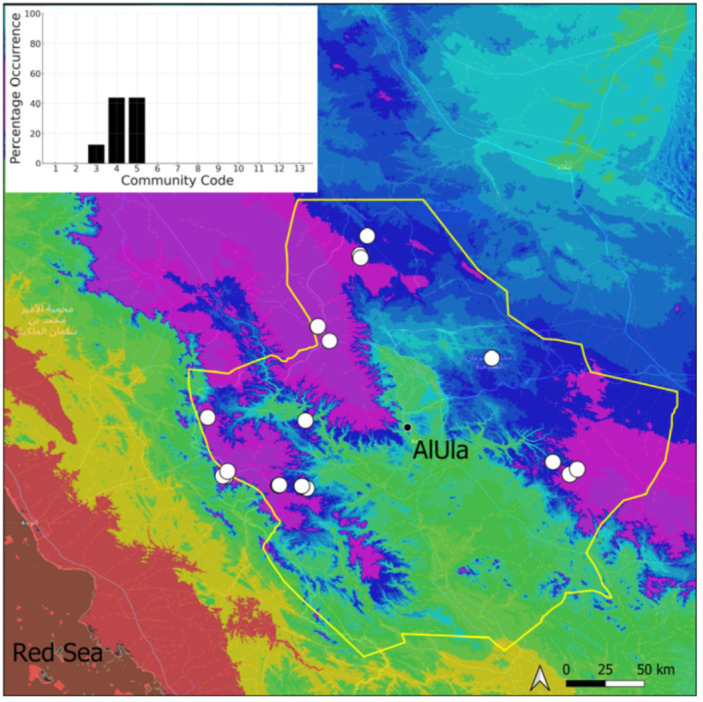
Distribution and habitats (inset) of *Grimmia orbicularis* Bruch ex Wilson in AlUla County.

**Figure 24 plants-14-00170-f024:**
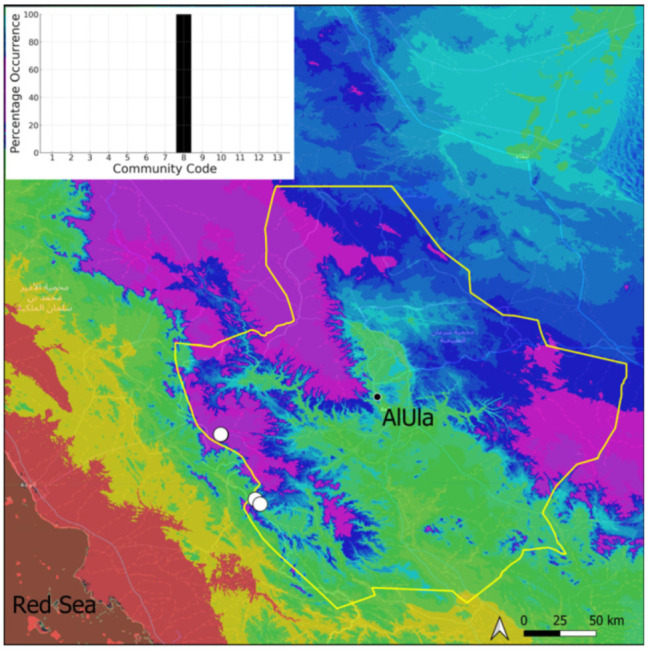
Distribution and habitat (inset) of *Gymnostomiella vernicosa* (Hook. ex Harv.) M. Fleisch. in AlUla County.

**Figure 25 plants-14-00170-f025:**
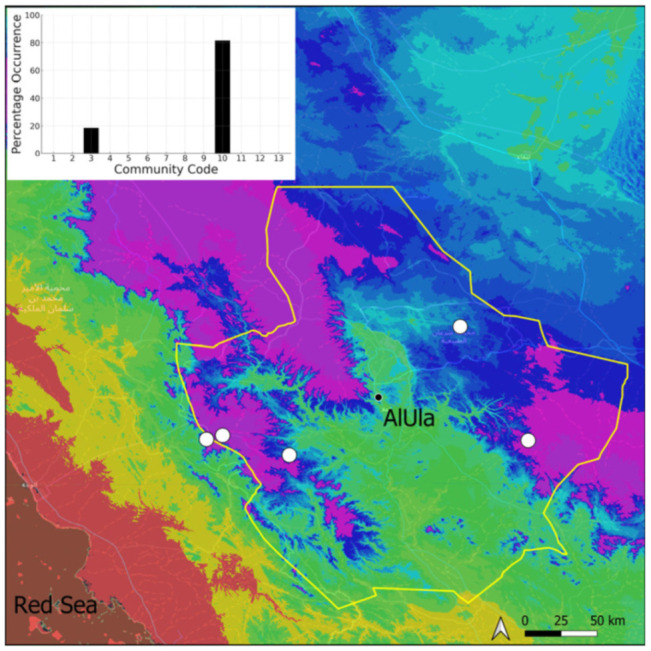
Distribution and habitats (inset) of *Gymnostomum calcareum* Nees & Hornsch. in AlUla County.

**Figure 26 plants-14-00170-f026:**
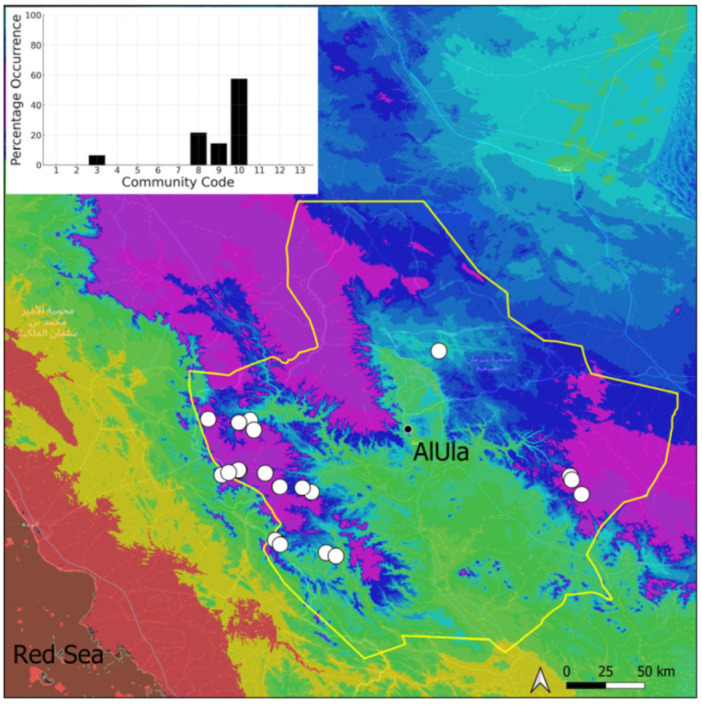
Distribution and habitats (inset) of *Gymnostomum mosis* (Lorentz) Jur. & Milde in AlUla County.

**Figure 27 plants-14-00170-f027:**
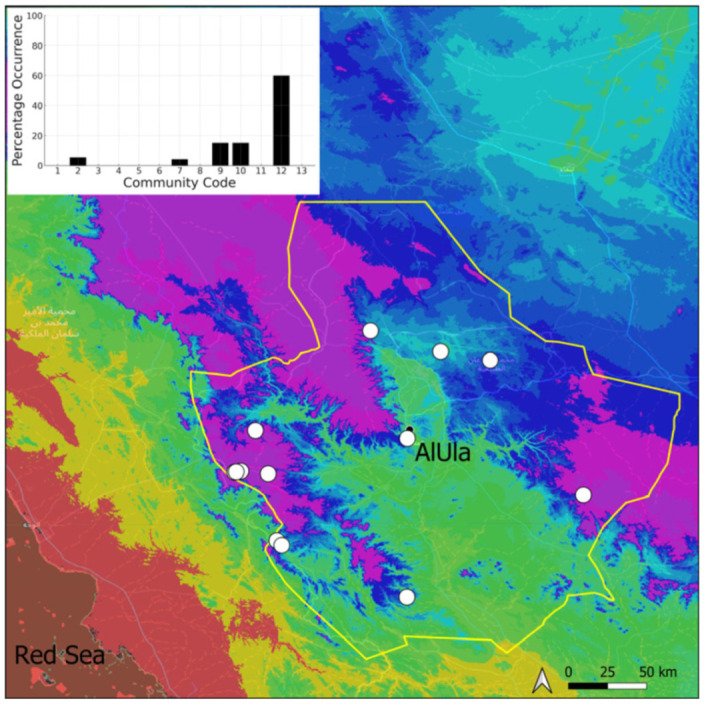
Distribution and habitats (inset) of *Gyroweisia tenuis* (Hedw.) Schimp. in AlUla County.

**Figure 28 plants-14-00170-f028:**
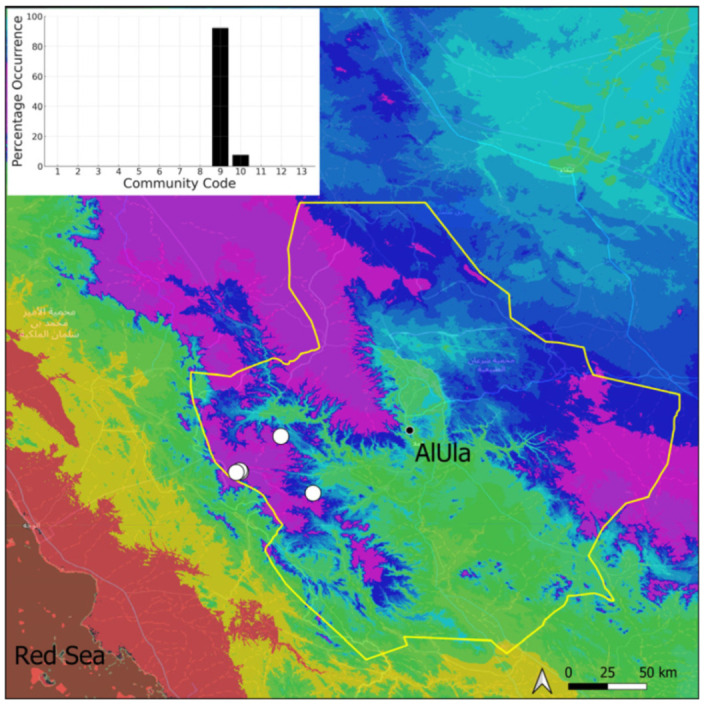
Distribution and habitats (inset) of *Hymenostylium hildebrandtii* (Müll. Hal.) R. H. Zander in AlUla County.

**Figure 29 plants-14-00170-f029:**
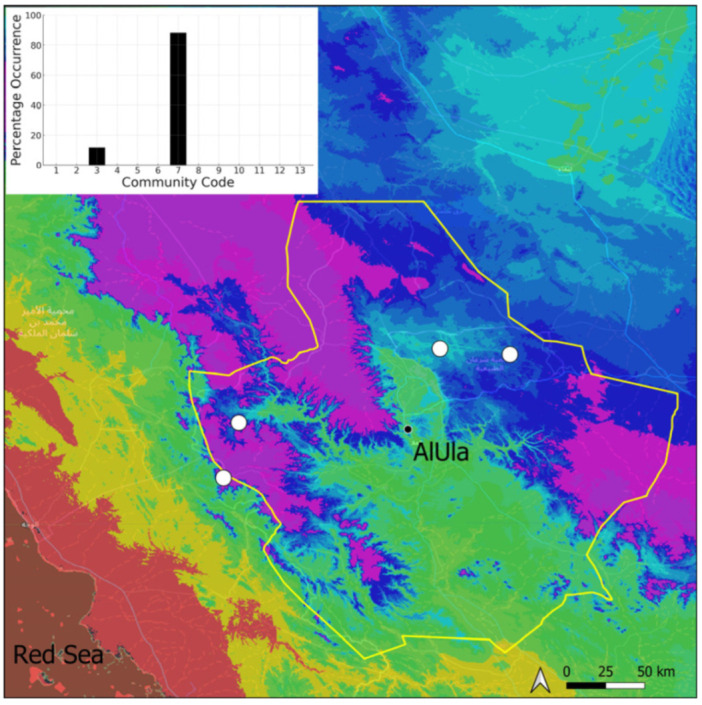
Distribution and habitats (inset) of *Microbryum davallianum* (Sm.) R. H. Zander in AlUla County.

**Figure 30 plants-14-00170-f030:**
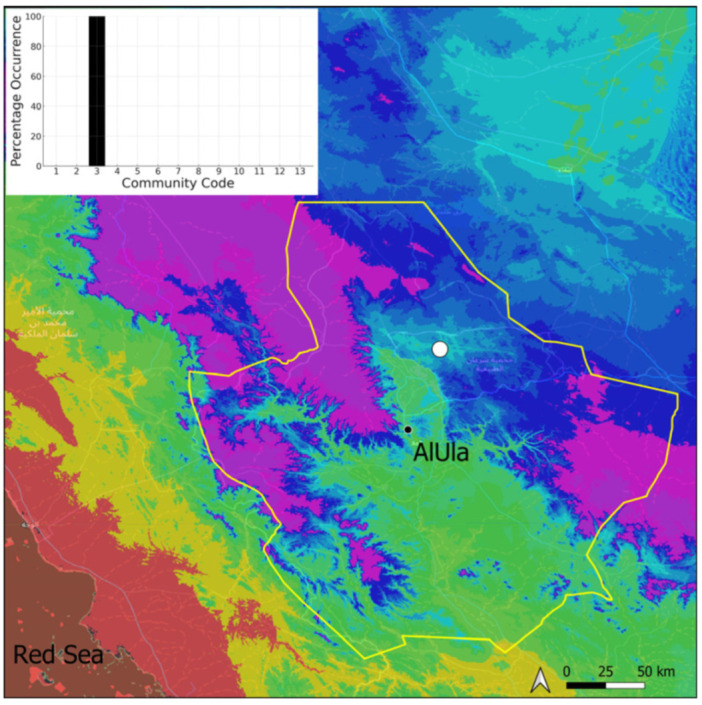
Distribution and habitat (inset) of *Microbryum rectum* (With.) R. H. Zander in AlUla County.

**Figure 31 plants-14-00170-f031:**
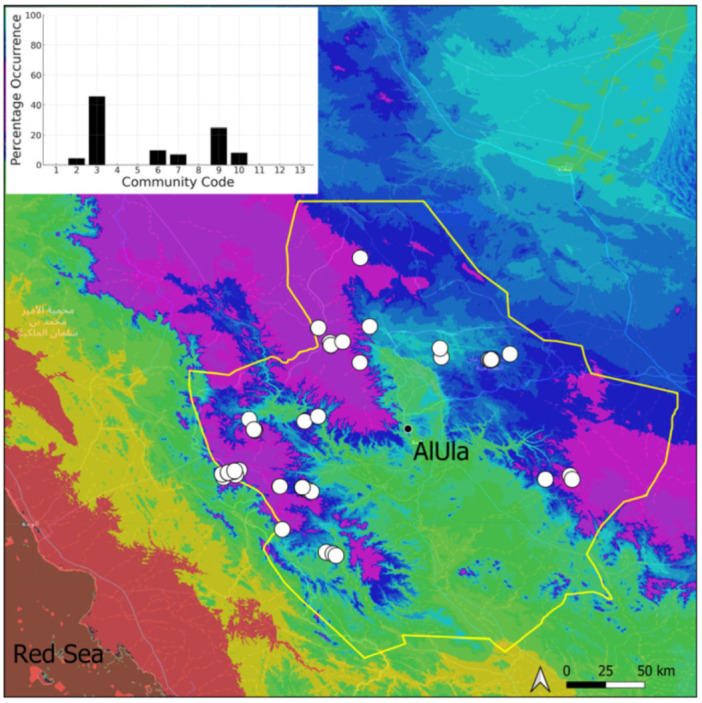
Distribution and habitats (inset) of *Microbryum starckeanum* (Hedw.) R. H. Zander in AlUla County.

**Figure 32 plants-14-00170-f032:**
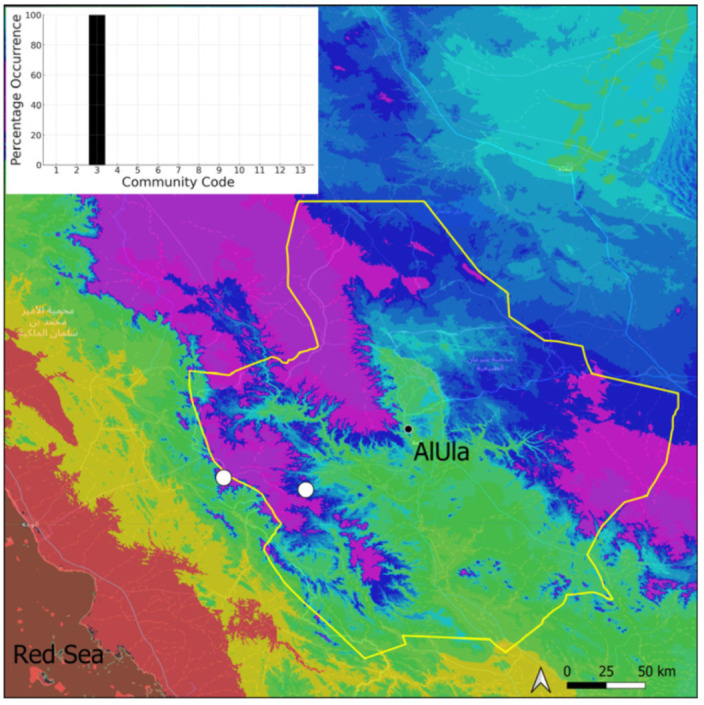
Distribution and habitat (inset) of *Molendoa handelii* (Schiffn.) Brinda & R. H. Zander in AlUla County.

**Figure 33 plants-14-00170-f033:**
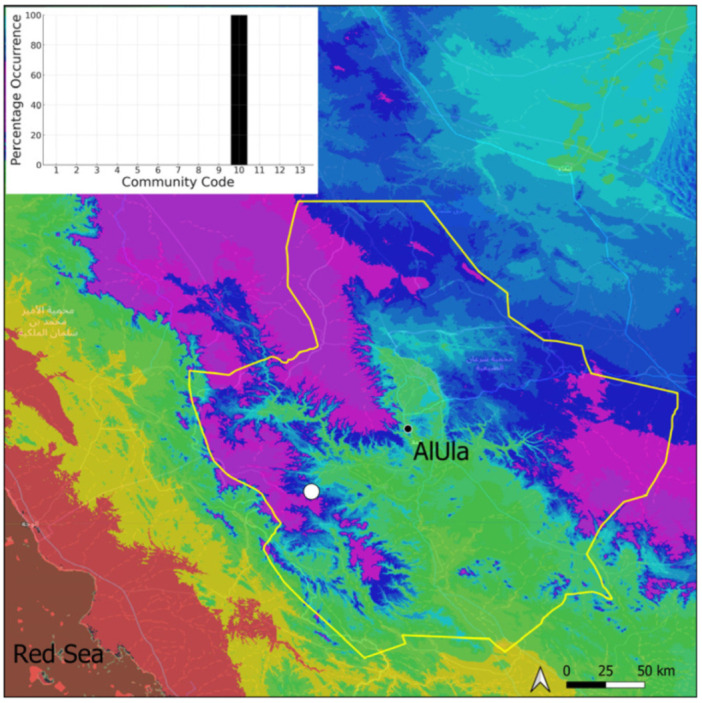
Distribution and habitat (inset) of *Plagiochasma rupestre* (J. R. Forst. & G. Forst.) Steph. in AlUla County.

**Figure 34 plants-14-00170-f034:**
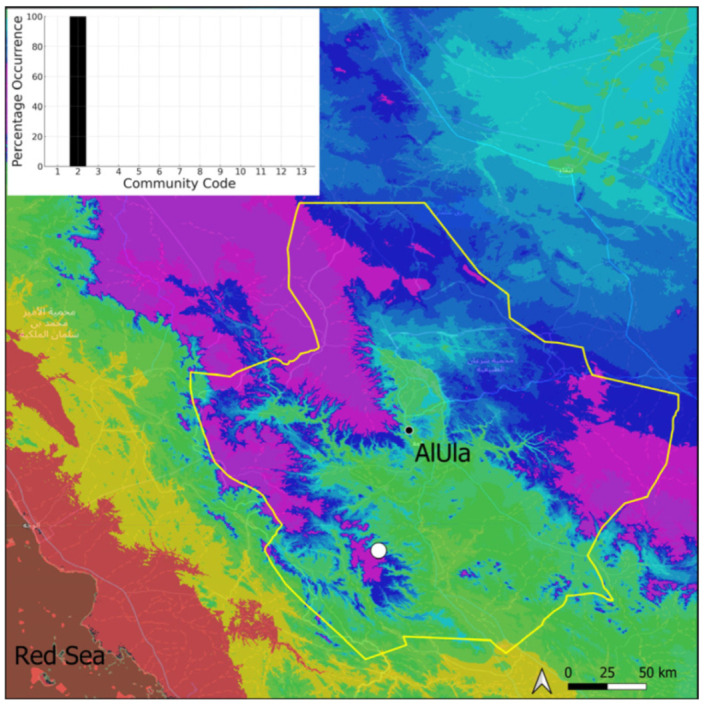
Distribution and habitat (inset) of *Pterygoneurum ovatum* (Hedw.) Dixon in AlUla County.

**Figure 35 plants-14-00170-f035:**
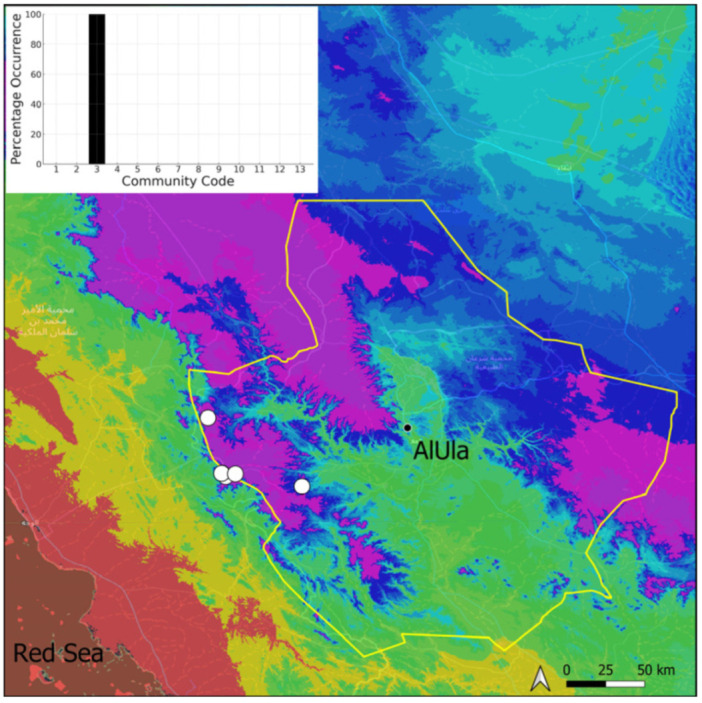
Distribution and habitat (inset) of *Ptychostomum capillare* (Hedw.) Holyoak & N. Pedersen in AlUla County.

**Figure 36 plants-14-00170-f036:**
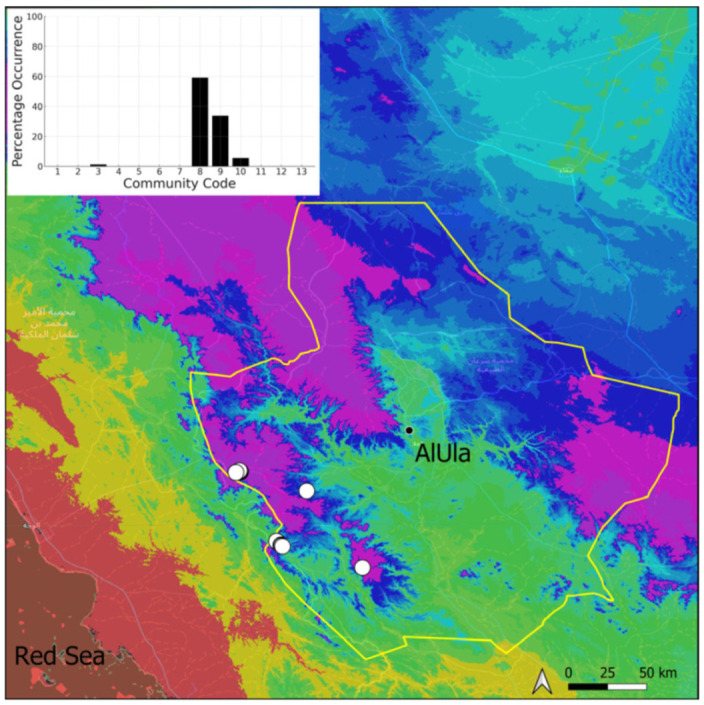
Distribution and habitats (inset) of *Ptychostomum cellulare* (Hook.) D. Bell & Holyoak in AlUla County.

**Figure 37 plants-14-00170-f037:**
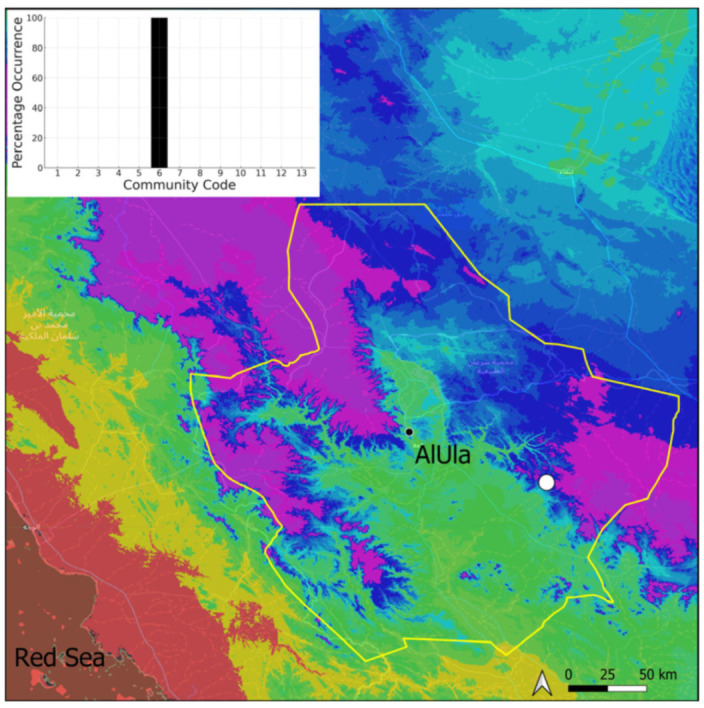
Distribution and habitat (inset) of *Ptychostomum pseudotriquetrum* (Hedw.) J. R. Spence & H. P. Ramsay in AlUla County.

**Figure 38 plants-14-00170-f038:**
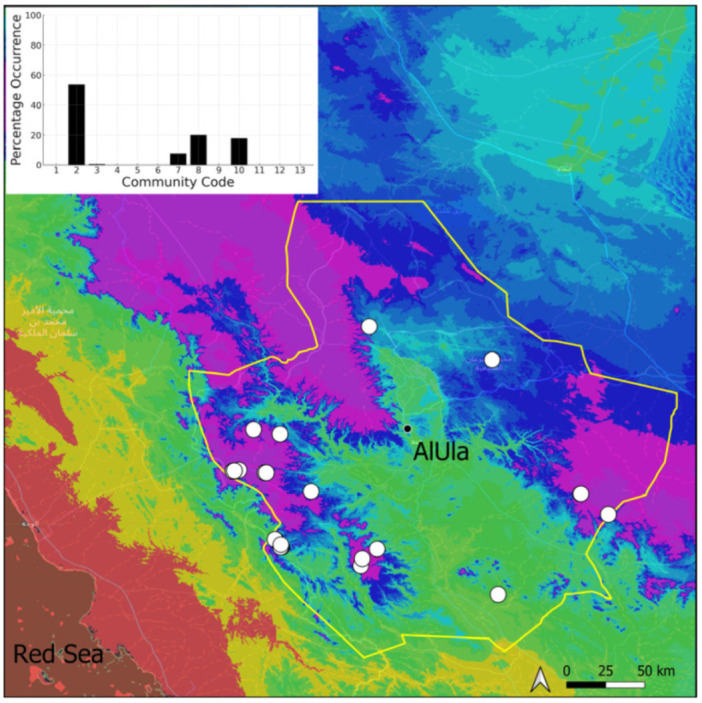
Distribution and habitats (inset) of *Riccia cavernosa* Hoffm. in AlUla County.

**Figure 39 plants-14-00170-f039:**
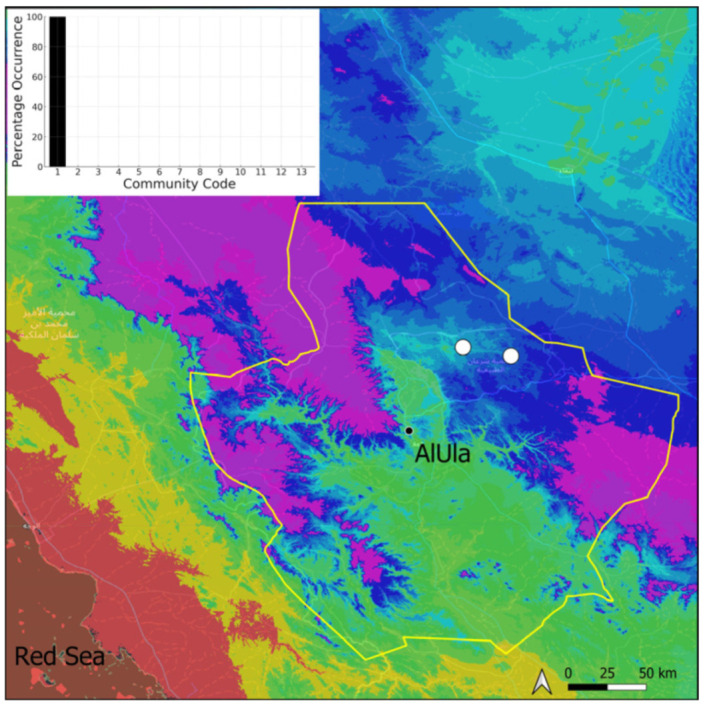
Distribution and habitat (inset) of *Riella affinis* M. Howe & Underw. in AlUla County.

**Figure 40 plants-14-00170-f040:**
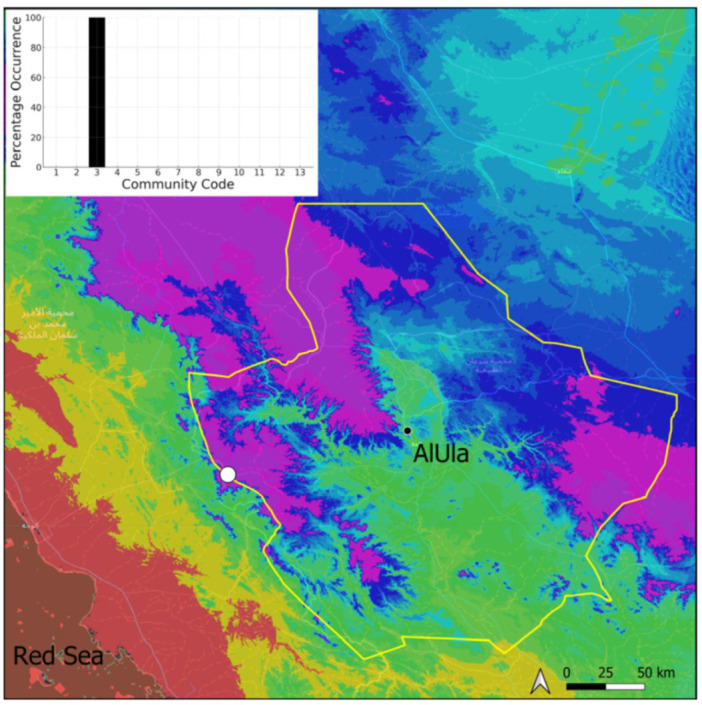
Distribution and habitat (inset) of *Syntrichia caninervis* Mitt. var. *caninervis* in AlUla County.

**Figure 41 plants-14-00170-f041:**
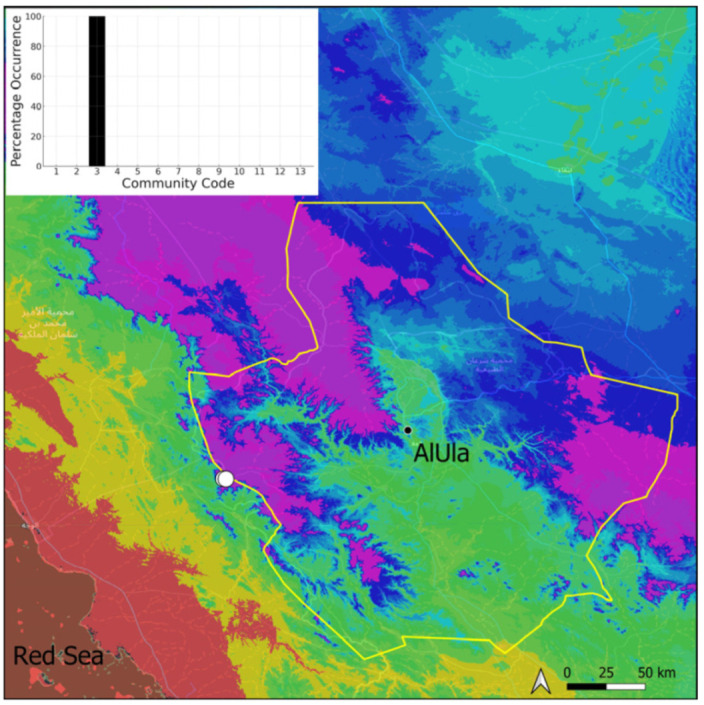
Distribution and habitat (inset) of *Syntrichia rigescens* (Broth. & Geh.) Ochyra in AlUla County.

**Figure 42 plants-14-00170-f042:**
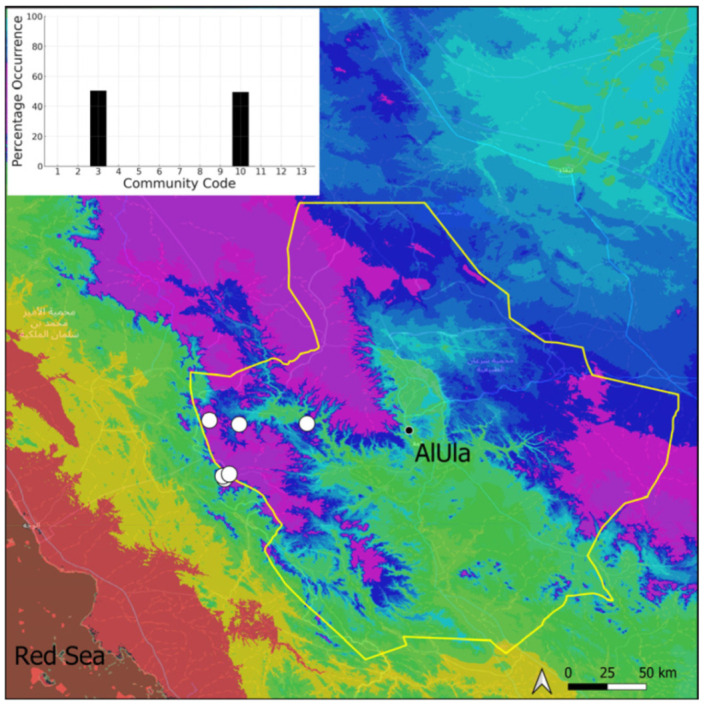
Distribution and habitats (inset) of *Targionia hypophylla* L. in AlUla County.

**Figure 43 plants-14-00170-f043:**
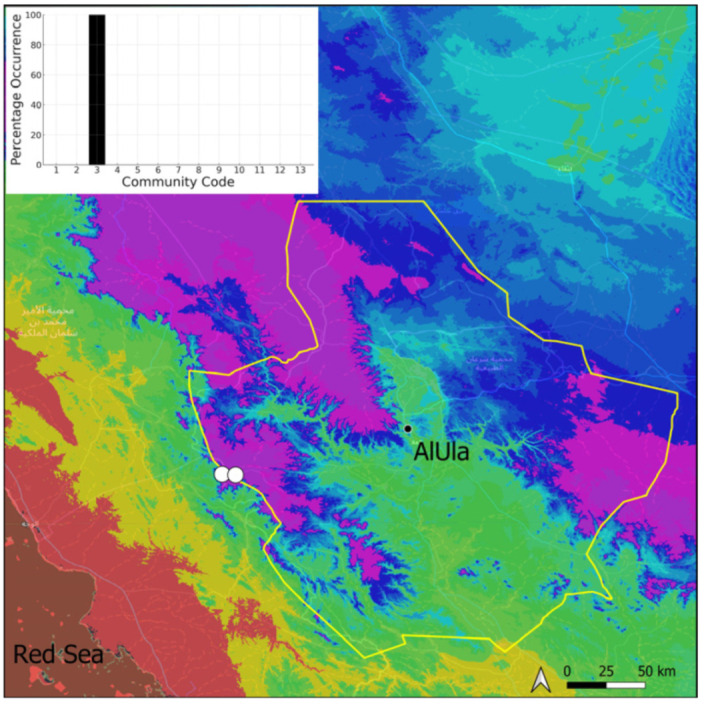
Distribution and habitat (inset) of *Targionia lorbeeriana* Müll. Frib. in AlUla County.

**Figure 44 plants-14-00170-f044:**
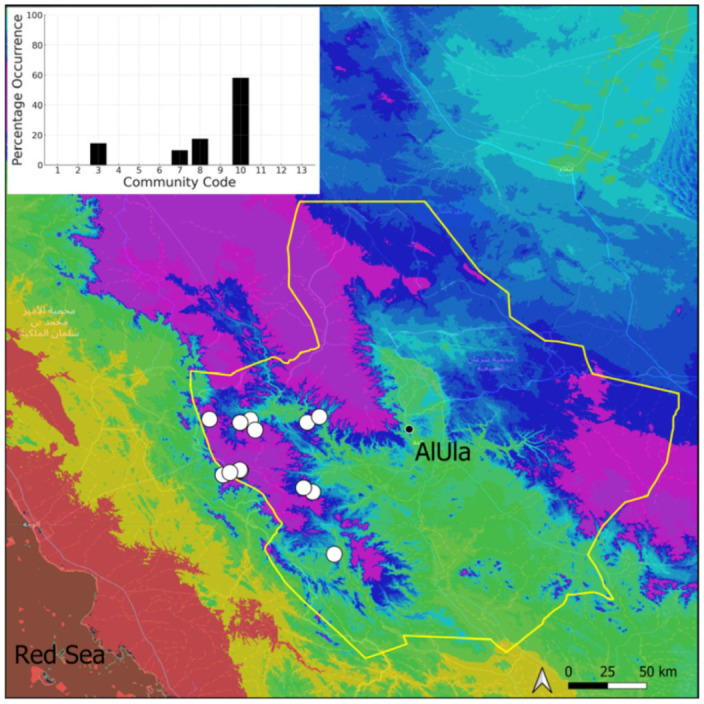
Distribution and habitats (inset) of *Timmiella barbuloides* (Brid.) Mönk. in AlUla County.

**Figure 45 plants-14-00170-f045:**
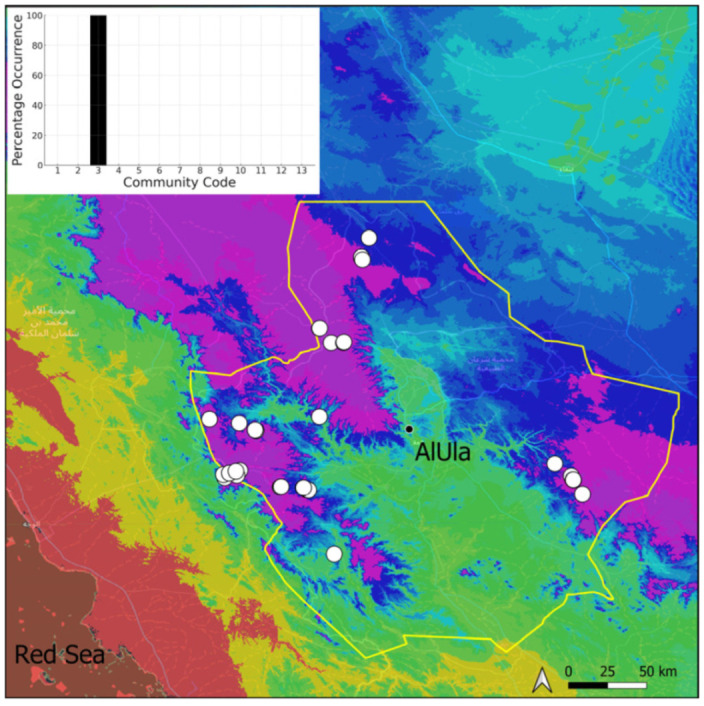
Distribution and habitat (inset) of *Tortula atrovirens* (Sm.) Lindb. in AlUla County.

**Figure 46 plants-14-00170-f046:**
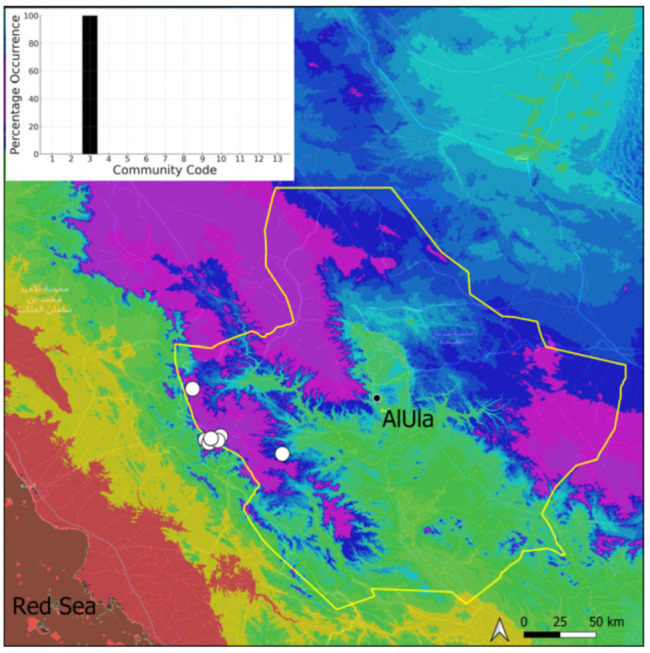
Distribution and habitat (inset) of *Tortula inermis* (Brid.) Mont. in AlUla County.

**Figure 47 plants-14-00170-f047:**
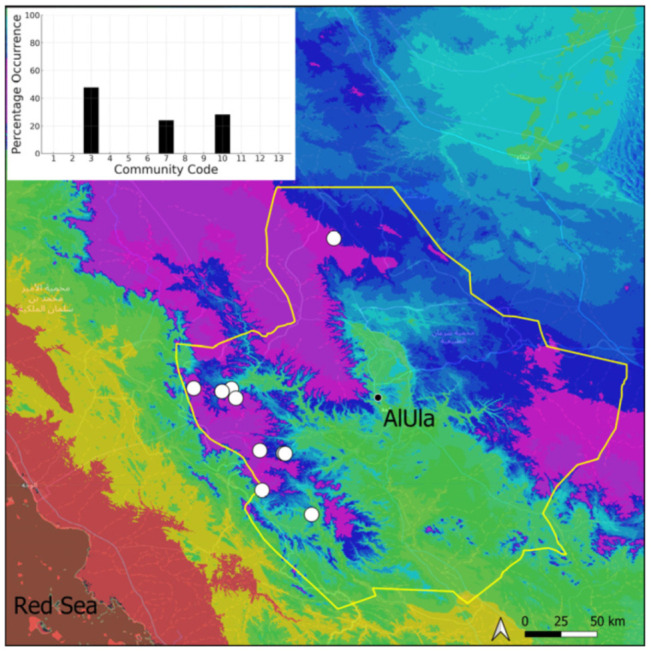
Distribution and habitats (inset) of *Tortula mucronifera* W. Frey, Kürschner & Ros in AlUla County.

**Figure 48 plants-14-00170-f048:**
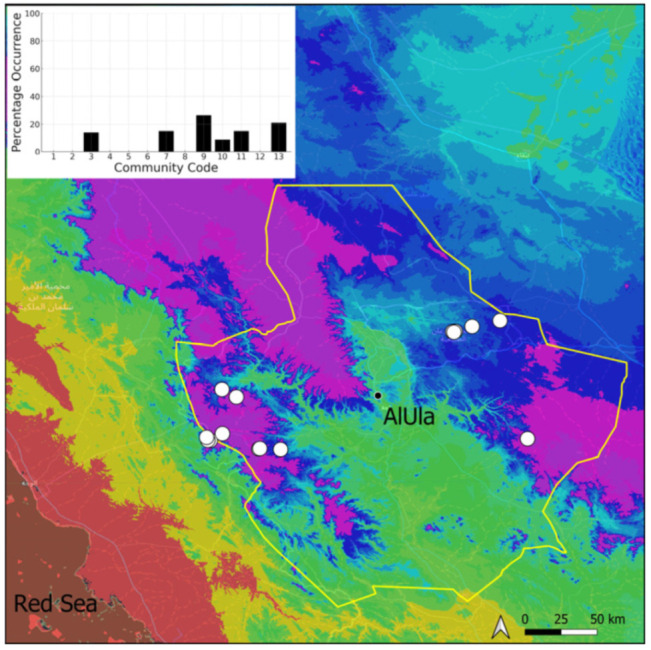
Distribution and habitats (inset) of *Tortula muralis* Hedw. in AlUla County.

**Figure 49 plants-14-00170-f049:**
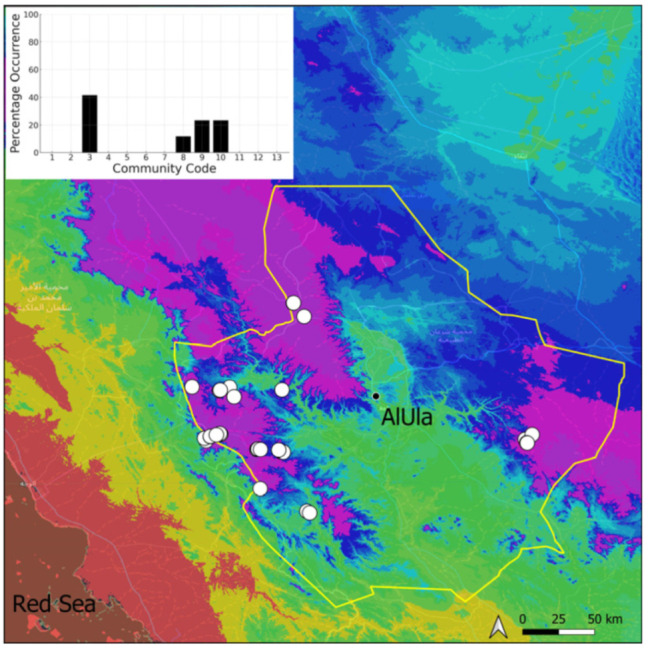
Distribution and habitats (inset) of *Trichostomopsis australasiae* (Hook. & Grev.) H. Rob. in AlUla County.

**Figure 50 plants-14-00170-f050:**
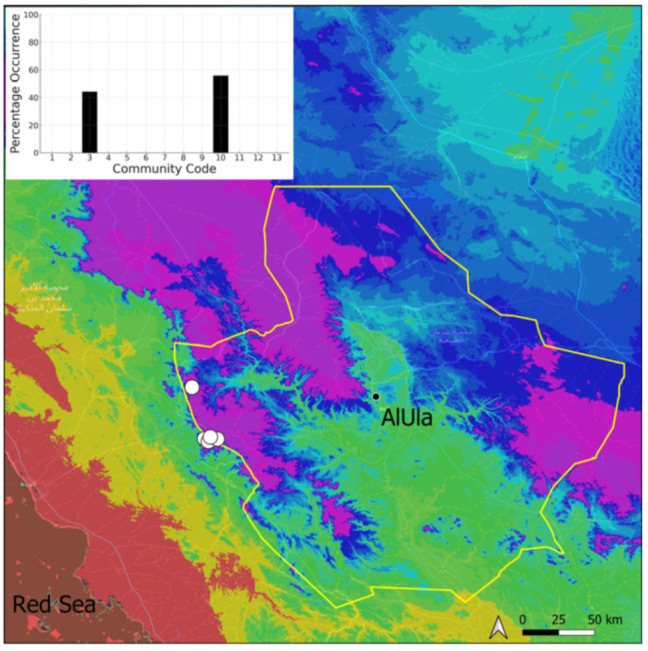
Distribution and habitats (inset) of *Vinealobryum vineale* (Brid.) R. H. Zander in AlUla County.

**Figure 51 plants-14-00170-f051:**
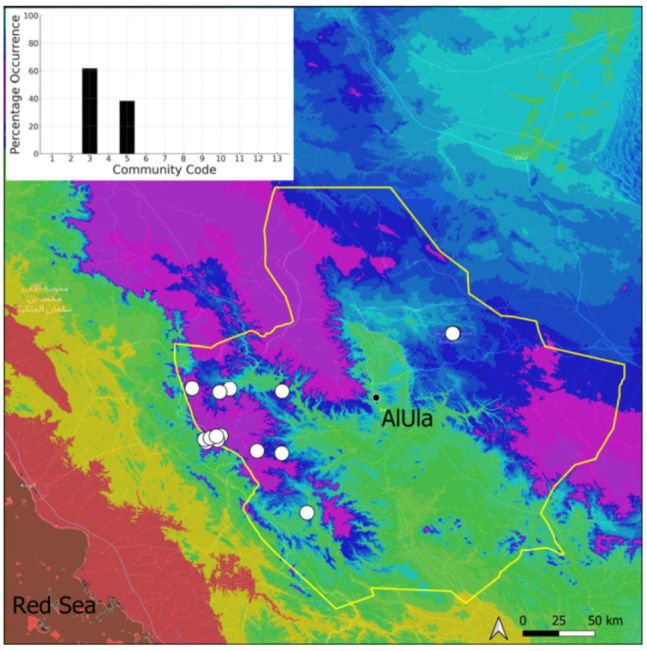
Distribution and habitats (inset) of *Weissia condensa* (Voit) Lindb. in AlUla County.

**Figure 52 plants-14-00170-f052:**
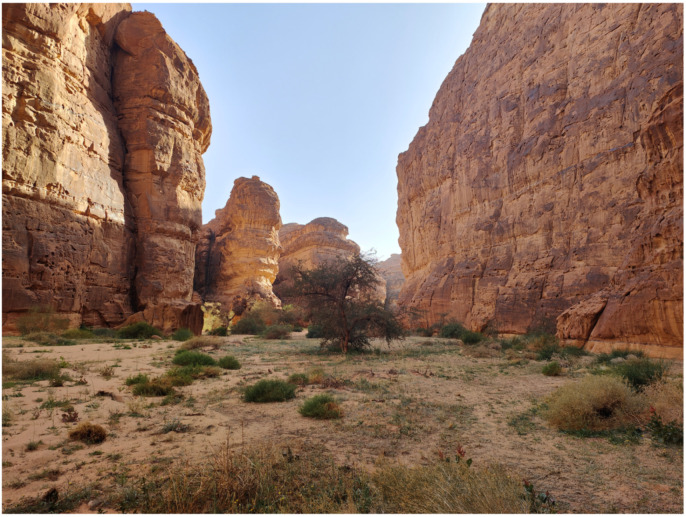
Giant labyrinth in the Sharaan Nature Reserve; bryophytes are confined to the base of the sandstone cliffs.

**Figure 53 plants-14-00170-f053:**
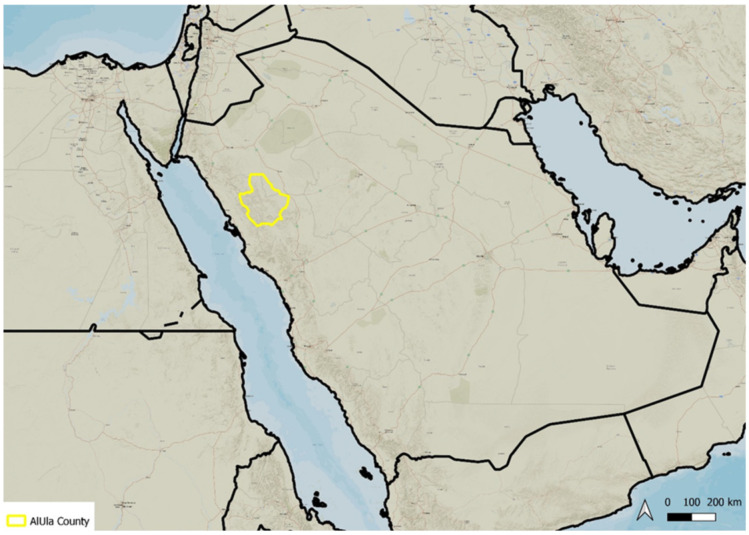
Location of AlUla County within the Arabian Peninsula.

**Figure 54 plants-14-00170-f054:**
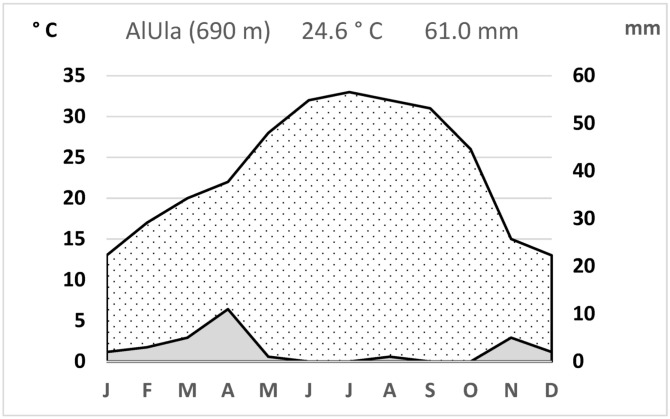
Climatogram for AlUla (26°37′ N; 37°58′ E) meteorological station (Data: 1991–2021).

**Figure 55 plants-14-00170-f055:**
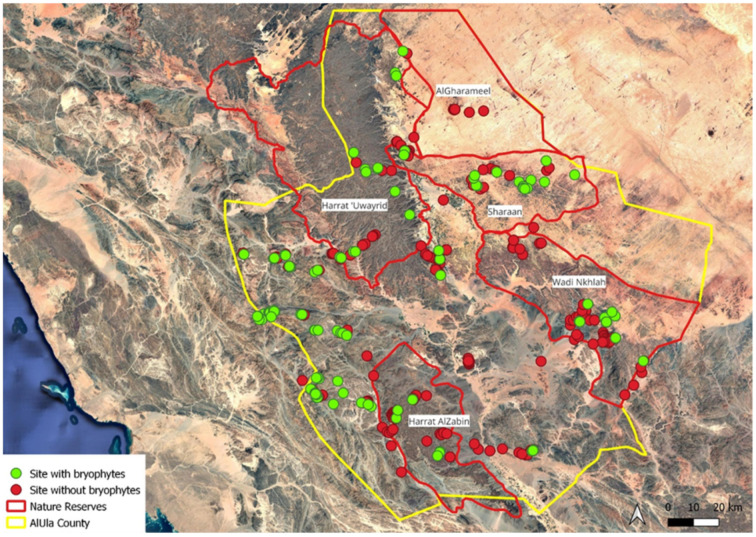
Location of surveyed sites (nature reserves are indicated with red lines).

**Table 1 plants-14-00170-t001:** Taxonomy and frequency of bryophyte taxa observed in AlUla County (see Section 5, Materials and Methods—C: relatively common species, probably often underestimated; R: relatively rare species; TR: seemingly very rare species and likely not underestimated; x: species with high conservational importance; xx: species with very high conservational importance; xxx: species with exceptional conservational importance; M: moss; L: liverwort).

Taxon	Number of Occurrences	Frequency in AlUla	Frequency on Arabian Peninsula	Heritage Value	Phylogeny	Family
*Acaulon triquetrum*	1	0.12	R	xx	M	Pottiaceae
*Aloina rigida*	4	0.48	C		M	Pottiaceae
*Bryum dichotomum*	33	3.92	C		M	Bryaceae
*Bryum* sp.	51	6.06			M	Bryaceae
*Clevea spathysii*	2	0.24	R	xx	L	Cleveaceae
*Crossidium aberrans* (Figure 1)	33	3.92	C		M	Pottiaceae
*Crossidium crassinervium*	36	4.28	C		M	Pottiaceae
*Crossidium deserti*	1	0.12	C		M	Pottiaceae
*Crossidium squamiferum*	30	3.56	C		M	Pottiaceae
*Didymodon desertorum*	22	2.61	C		M	Pottiaceae
*Encalypta vulgaris*	1	0.12	C		M	Encalyptaceae
*Entosthodon* cf. *commutatus*	42	4.99			M	Funariaceae
*Entosthodon duriaei*	17	2.02	C		M	Funariaceae
*Entosthodon muhlenbergii*	7	0.83	C		M	Funariaceae
*Entosthodon* sp.	14	1.65			M	Funariaceae
*Eucladium verticillatum*	11	1.31	C	x	M	Pottiaceae
*Fissidens* cf. *arnoldii*	42	4.99	C		M	Fissidentaceae
*Funaria hygrometrica*	18	2.14	C		M	Funariaceae
*Geheebia siccula*	3	0.36	R	x	M	Pottiaceae
*Geheebia tophacea*	39	4.63	C		M	Pottiaceae
*Grimmia anodon*	2	0.24	R	xx	M	Grimmiaceae
*Grimmia orbicularis*	41	4.87	C		M	Grimmiaceae
*Gymnostomiella vernicosa*	5	0.59	R	xxx	M	Pottiaceae
*Gymnostomum calcareum*	6	0.71	C		M	Pottiaceae
*Gymnostomum mosis* (Figure 2)	28	3.33	C		M	Pottiaceae
*Gyroweisia tenuis*	13	1.54	C		M	Pottiaceae
*Hymenostylium hildebrandtii*	5	0.59	R	xxx	M	Pottiaceae
*Microbryum davallianum*	4	0.48	C		M	Pottiaceae
*Microbryum rectum*	1	0.12	R	xx	M	Pottiaceae
*Microbryum starckeanum*	54	6.41	C		M	Pottiaceae
*Molendoa handelii*	2	0.24	R	xx	M	Pottiaceae
*Plagiochasma rupestre*	1	0.12	C		L	Aytoniaceae
*Pterygoneurum ovatum*	1	0.12	C		M	Pottiaceae
*Ptychostomum capillare*	7	0.83	C		M	Bryaceae
*Ptychostomum cellulare*	12	1.43	TR	xxx	M	Bryaceae
*Ptychostomum pseudotriquetrum*	2	0.24	TR	xx	M	Bryaceae
*Riccia cavernosa*	32	3.80	C		L	Ricciaceae
*Riella affinis*	2	0.24	TR	xxx	L	Riellaceae
*Syntrichia caninervis* var. *caninervis*	2	0.24	C		M	Pottiaceae
*Syntrichia rigescens*	3	0.36	R	x	M	Pottiaceae
*Targionia hypophylla*	10	1.19	C		L	Targioniaceae
*Targionia lorbeeriana*	2	0.24	C		L	Targioniaceae
*Timmiella barbuloides*	18	2.14	C		M	Pottiaceae
*Tortula atrovirens*	48	5.70	C		M	Pottiaceae
*Tortula inermis*	17	2.02	C		M	Pottiaceae
*Tortula mucronifera*	17	2.02	C		M	Pottiaceae
*Tortula muralis*	20	2.38	C		M	Pottiaceae
*Trichostomopsis australasiae*	52	6.18	C		M	Pottiaceae
*Vinealobryum vineale*	8	0.95	C		M	Pottiaceae
*Weissia condensa*	20	2.38	C		M	Pottiaceae

**Table 2 plants-14-00170-t002:** Number of sites hosting mixed populations of species from the genus *Crossidium.*

	*Crossidium* *aberrans*	*Crossidium* *crassinervium*	*Crossidium deserti*	*Crossidium squamiferum*
*Crossidium aberrans*		4		3
*Crossidium crassinervium*	4			8
*Crossidium deserti*				1
*Crossidium squamiferum*	3	8	1	

**Table 3 plants-14-00170-t003:** Endemic taxa of the Arabian Peninsula and their current taxonomic status (in bold font: taxa currently considered as endemic to the Arabian Peninsula; the two species recorded in AlUla County are highlighted).

Accepted Nomenclature	Description Name	Family	Global Range	Diagnosis Reference
***Bryum nanoapiculatum* Ochi & Kürschner**	*Bryum nanoapiculatum* Ochi & Kürschner	Bryaceae	Oman; Yemen	[29]
*Crossidium davidai* Catches.	*Crossidium asirense* W. Frey & Kürschner	Pottiaceae	Australia; Arabian Peninsula	[14]
***Crossidium deserti* W. Frey & Kürschner**	*Crossidium deserti* W. Frey & Kürschner	Pottiaceae	Saudi Arabia	[13]
*Crossidium laxefilamentosum* W. Frey & Kürschner	*Crossidium laxefilamentosum* W. Frey & Kürschner	Pottiaceae	Arabian Peninsula; North Africa	[22]
***Crossidium woodii* (Delgad.) R. H. Zander**	*Pseudaloina woodii* Delgad.	Pottiaceae	Yemen	[30]
***Fissidens laxetexturatus* Brugg.-Nann.**	*Fissidens laxetexturatus* Brugg.-Nann.	Fissidentaceae	Oman, Yemen	[16]
*Fissidens sciophyllus* Mitt.	*Fissidens arabicus* Pursell & Kürschner	Fissidentaceae	Tropical Africa, and Arabian Peninsula	[31]
***Riccia crenatodentata* O. H. Volk**	*Riccia crenatodentata* O. H. VOLK	Ricciaceae	Arabian Peninsula;	[32]
***Schlotheimia balfourii* Mitt.**	*Schlotheimia balfourii* Mitt.	Orthotrichaceae	Yemen (Socotra)	[33]
***Sematophyllum socotrense* W. R. Buck**	*Sematophyllum socotrense* W. R. Buck	Sematophyllaceae	Yemen (Socotra)	[34]
*Splachnobryum aquaticum* Müll. Hal.	*Splachnobryum arabicum* Dixon	Splachnobryaceae	Asia, Africa, and Arabian Peninsula	[35]
***Targionia hypophylla* subsp. *linealis* W. Frey & Kürschner**	*Targionia hypophylla* subsp. *linealis* W. Frey & Kürschner	Targioniaceae	Arabian Peninsula	[27]
*Tortella humilis* (Hedw.) Jenn.	*Barbula schweinfurthiana* Müll. Hal.	Pottiaceae	Sub-cosmopolitan	[36]
***Tortella smithii* C. C. Towns.**	*Tortella smithii* C. C. Towns.	Pottiaceae	Yemen (including Socotra)	[37]
***Tortula mucronifera* W. Frey, Kürschner & Ros**	*Tortula mucronifera* W. Frey, Kürschner & Ros	Pottiaceae	Arabian Peninsula and Jordan	[38]
***Weissia artocosana* R. H.** **Zander**	*Weissia artocosana* R. H. Zander	Pottiaceae	Yemen (Socotra)	[39]
***Weissia socotrana* Mitt.**	*Weissia socotrana* Mitt.	Pottiaceae	Yemen (Socotra)	[33]

**Table 4 plants-14-00170-t004:** AlUla County bryophytic communities [70].

Community Code	Ecology	Community Name	Description
1	Aquatic community	*Riella affinis* community	Monospecific, ephemeral community of shallow water bodies
6	Tufa communities	*Eucladio verticillatae–Adiantetum capillusi-veneris* Br.-BI. ex Horvatić 1934	Community of active flowing tufa deposits
7		*Geheebia tophacea* community	Community of periodically drying tufa deposits
8		*Gymnostomiella vernicosa* community	Community of tufa deposits with tropical affinities
9		*Hymenostylium hildebrandtii* community	Community of tufa deposits in the minor beds of small temporary wadi
10		*Gymnostomum mosis* community	Community of periodically drying tufa deposits
12		*Gyroweisia tenuis* community	Community of periodically drying tufa deposits
13		*Entosthodon duriaei* community	Community of periodically drying tufa deposits
2	Terricolous communities	*Riccia cavernosa* community	Community of sandy shores of temporary wadis
3		*Pottiaceae* community	Pottiaceae-rich community of scree and the base of rock walls
11		*Bryum sp.* community	Community of compacted, temporarily moist soils
4	Saxicolous communities	*Grimmia anodon* community	Community of high-altitude granite boulders
5		*Grimmia orbicularis* community	Community of granite or sandstone boulders and rock walls

**Table 5 plants-14-00170-t005:** Nomenclature for names deviating from [5].

Current Nomenclature	Kürschner & Frey, 2020	Followed Reference
*Molendoa handelii* (Schiffn.) Brinda & R. H. Zander	*Anoectangium handelii* Schiffn.	[71]
*Clevea spathysii* (Lindenb.) Müll. Frib.	*Athalamia spathysii* (Lindenb.) S. Hatt.	[72]
*Ptychostomum capillare* (Hedw.) Holyoak & N. Pedersen	*Bryum capillare* Hedw.	[12]
*Ptychostomum pseudotriquetrum* (Hedw.) J. R. Spence & H. P. Ramsay	*Bryum pseudotriquetrum* (Hedw.) G. Gaertn.	[12]
*Ptychostomum torquescens* (Bruch & Schimp.) Ros & Mazimpaka	*Bryum torquescens* Bruch & Schimp.	[12]
*Trichostomopsis australasiae* (Hook. & Grev.) H. Rob.	*Didymodon australasiae* (Hook. & Grev.) R. H. Zander	[17]
*Geheebia siccula* (M.J. Cano, Ros, García-Zam. & J. Guerra) J. A. Jiménez & M. J. Cano	*Didymodon sicculus* M. J. Cano, Ros, García-Zam. & J. Guerra	[17]
*Geheebia tophacea* (Brid.) R. H. Zander	*Didymodon tophaceus* (Brid.) Lisa	[17]
*Vinealobryum vineale* (Brid.) R. H. Zander	*Didymodon vinealis* (Brid.) R. H. Zander	[17]
*Riella affinis* M. Howe & Underw.	not mentioned	[73]
*Entosthodon commutatus* Durieu & Mont.	not mentioned	[74]
*Didymodon desertorum* (J. Froehl.) J. A. Jiménez & M. J. Cano	not recognized	[9]
*Ptychostomum cellulare* (Hook.) D. Bell & Holyoak	*Plagiobryoides cellularis* (Hook.) J. R. Spence	[12]

## Data Availability

The datasets presented in this article are not readily available because of the data ownership. Requests to access the datasets should be directed to the Vice President of Wildlife & Natural Heritage, RCU.

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
