# Peer review of "Bryophyte Flora of AlUla County (Saudi Arabia)—Distribution, Ecology, and Conservation"

_plants, 2025, doi:10.3390/plants14020170_

Round 1

Reviewer 1 Report

Comments and Suggestions for Authors

The manuscript is essentially valuable for the knowledge of the bryophyte diversity of the understudied Arabian Peninsula. The manuscript is a contribution to the knowledge on bryophyte flora of Saudi Arabia as well as on ecological preferences of recorded taxa in such an extreme environmental conditions, unfavourable for most of the plants.

I have put my comments and corrections directly within the manuscript. However, I emphasize several things which should be modified:

1. I strongly advise to joint several maps into one larger picture, e.g. 2 or even 3 maps could fit together within the limits of the A4 paper size, with using colour spectrum showing altitudes only once on the right side of that composed picture. Moreover, 9 or 12 maps could be in composition at one A4 paper as a unique picture, for example. 

2. The font of the text in all tables and pictures should be enlarged, since it is illegible.

3. Please rewrite the sentences in which you mentione submitted manuscripts or manuscripts in preparations (Hugonnot, V.; Pépin, F.; Freedman, J. New Bryophyte Species for Saudi Arabia and the Arabian Peninsula from AlUla County. 812 Nova Hedwigia, submitted: Hugonnot, V.; Pépin, F.; Freedman, J. Bryophyte Communities from AlUla County (Saudi Arabia). In Preparation, 2024.). Those references could not be cited as "published" as it has been stated in many places within the manuscript.

Author Response

  1. I strongly advise to joint several maps into one larger picture, e.g. 2 or even 3 maps could fit together within the limits of the A4 paper size, with using colour spectrum showing altitudes only once on the right side of that composed picture. Moreover, 9 or 12 maps could be in composition at one A4 paper as a unique picture, for example.

OK, we have reorganized the plates in the entire MS.

  1. The font of the text in all tables and pictures should be enlarged, since it is illegible.

OK, we have enlarged the plates.

  1. Please rewrite the sentences in which you mentione submitted manuscripts or manuscripts in preparations (Hugonnot, V.; Pépin, F.; Freedman, J. New Bryophyte Species for Saudi Arabia and the Arabian Peninsula from AlUla County. 812 Nova Hedwigia, submitted: Hugonnot, V.; Pépin, F.; Freedman, J. Bryophyte Communities from AlUla County (Saudi Arabia). In Preparation, 2024.). Those references could not be cited as "published" as it has been stated in many places within the manuscript.

OK, we have modified the formulation each time [11] (our publication currently under review) was mentioned.

Additionally, we have modified each of the suggested formal changes in the MS (mostly italics and typographic errors)

we also have added photographs, to fill the empty space

thanks very much for the referee's comments, that allowed to improve our MS

Reviewer 2 Report

Comments and Suggestions for Authors

The colors in Fig. 2 are not clearly identified as to what they mean.Maybe you don't mean "ecology" but rather habitat. Ecology is a process-based concept. Give a short explanation at least with Fig. 2.  You might put before the color maps the general map of KSA Fig. 60 with an inset showing where in nw KSA the area studied is. It is too bad that this journal puts methods near the end. Congratulations to the authors for recognizing that morphology is more important than molecular study, as with Didymodon tophaceus group.

"Accepted nomenclature"? Accepted by whom? If you are referring to the World Flora Online, this is a cladist-curated list and the accepted names are often not generally accepted except by cladists.

This is an important paper because the species are highly habitat-specific and can be used to infer past distributions based on climate change. The paper is very well done!

Author Response

The colors in Fig. 2 are not clearly identified as to what they mean.Maybe you don't mean "ecology" but rather habitat. Ecology is a process-based concept.

OK, we have changed “ecology” to “habitat”

Give a short explanation at least with Fig. 2.  You might put before the color maps the general map of KSA Fig. 60 with an inset showing where in nw KSA the area studied is.

It is not possible to repeat the information provided in the M&M section unfortunately.

"Accepted nomenclature"? Accepted by whom? If you are referring to the World Flora Online, this is a cladist-curated list and the accepted names are often not generally accepted except by cladists.

Ok we have changed : current nomenclature. We follow the most up-to-date taxonomic treatments, and mostly Kürschner & Frey.

Reviewer 3 Report

Comments and Suggestions for Authors

just modify the position of "Figure X" not near the title of each section/moss but in the text.

for the rest, very good job!

Author Response

just modify the position of "Figure X" not near the title of each section/moss but in the text.

Ok we have moved this information in the body of the text